



**An Analysis of Conflict and Cooperation Dynamics over Water Events in the Lancang-**
**Mekong River Basin**
Jing Wei[1], Yongping Wei[2], Fuqiang Tian[1*], Natalie Nott[2], Claire de Witt[2], Liying Guo[1], You Lu[1]
[1]Department of Hydraulic Engineering, State Key Laboratory of Hydroscience and Engineering,
Tsinghua University, Beijing 100084, China
[2] School of Earth and Environmental Sciences, University of Queensland, St. Lucia, QLD 4072,
Australia
Correspondence to:
Fuqiang Tian, tianfq@mail.tsinghua.edu.cn
Submitted to *Hydrology and Earth System Sciences*
Special issue: Socio-hydrology and Transboundary Rivers


**Abstract**
Riparian countries have their respective values and priorities for water management, and
their values of shared water often has possible impacts for their propensity to involve in
cooperative management and adhere to treaties/agreements. Improving transboundary
water management therefore firstly requires nuance understanding of the changing values
and interests of each riparian country to better understand factors that encourage and
discourage changes toward cooperation or conflict. This paper provides understanding of
the evolution of conflict and cooperation dynamics in Lancang-Mekong River Basin with
in-depth analysis of the perspectives of multiple countries. Newspaper articles were used
as a key data source as it provides insights into events reported on by the media that are
representative of each country/sector they are published within. The results depict a
continual trend of cooperative sentiments towards water events occurring within the region.
The six riparian states have had a greater average sentiment score for cooperation than
international countries for the majority of the study period showing that the region
perceived transboundary water management more positively than global audiences. Except
for few outliers, the trend also shows that countries further downstream showed lower
cooperative sentiments. Dam infrastructure was often negatively reported, thus, it is likely
a major contributor to conflict for the Lancang-Mekong River Basin, while events that are
positively reported are those that aid in connecting leaders and project developers between
riparian countries including meetings, bilateral and multilateral cooperation and
development projects. These findings provide the basis for further revealing the mechanism
of cooperation and conflicts through understanding these inherent and diverse perspectives
of each riparian country, we can gain an insight into the underlying interests that create
conflictive or cooperative environments and ultimately predict and manage
cooperation/conflict in transboundary rivers.
Keywords: transboundary river management, conflict and cooperation, Lancang-Mekong
river basin, newspaper, sentiment analysis, societal value, big data



## 1.  Introduction


Globally there are 310 transboundary rivers that flow across more than 47% of the world's land
surface (McCracken and Wolf, 2019), providing approximately 60% of the world's freshwater
(Wolf et al., 2005). Transboundary river flows across political boundaries with spatial and
temporal variance, often resulting in conflicting criteria for its uses among riparian nations. The
very different views on how the water should be used, and how it should be managed makes
collaborative management difficult (Sunchindah, 2013). Tensions and uncertainties often occur
when sharing this consumable, indispensable resource and compounded by the dynamic
interaction of hydrological, technical and social systems (Zeitoun and Mirumachi, 2008).
Transboundary rivers are therefore characterized for evolving cooperation and conflict dynamics
(Wolf et al., 1999;Petersen‐Perlman and Wolf, 2015;Yoffe et al., 2003;Zeitoun and Mirumachi,
2008). Given the future projections of population growth and aridity, some scholars even
proposed the idea of global 'water wars' (Cooley, 1984;Starr, 1991;Bulloch, 1995;Remans,
1995;Gleick, 1993), emphasizing that management of international rivers will be a challenging
task if knowledge of conflict and cooperation is not fully developed (Song and Whittington,
2004;Barnaby, 2009).
Understanding transboundary waters by conflict and cooperation has been a dominant approach
embraced by many scholars in different disciplines (Wolf et al., 2003;Yoffe et al., 2003;De
Stefano et al., 2010;Zawahri, 2008;Gleick, 1998). A large and growing body of literature has
attempted to explore factors that are potentially conductive to conflict, considering issues such as
water scarcity (Dinar, 2009), climate change (Gleditsch, 2012;Nordås and Gleditsch,
2007;Raleigh and Kniveton, 2012), water quality (Wolf et al., 2005), and the role of
transboundary treaties/river basin organizations (Song and Whittington, 2004;Dinar et al.,
2019;Berardo and Gerlak, 2012;Zawahri and Mitchell, 2011); while others have explored
cooperation management, focusing on scenario-based analysis of the distribution of benefits from
cooperation, and benefit-sharing mechanisms as pivotal role in motivating cooperation (Hogarth
and Dinar, 2015;Madani, 2010). Recently, conflict and cooperative dynamics in transboundary





rivers have been considered as a socio-hydrological phenomenon (Di Baldassarre et al., 2019),
emerged as a result of the long-term evolution of hydrological, political, economic, technical and
social processes settled within the transboundary system (Di Baldassarre et al., 2019). Socio-
hydrological approach is thus proposed in understanding transboundary river problem to unravel
how and why different actors came into cooperation.
As the first and fundamental advance to analyse the tendency for conflict or cooperation along
international rivers, few inventories have been built up to provide global snapshot of conflict and
cooperation dynamic to recognise future tensions. Often cited is the Transboundary Freshwater
Dispute Database (TFDD) developed by Oregon State University (Wolf, 1999) that compiled
historical water incidents, both conflictive and cooperative, on a global scale from 1948. Based
on the data, Basin in Risk Projects (BAR) (Yoffe and Larson 2001) categorised intensities of
water incidents, varying between-7 and +7, in order to understand possible social - political
threats. Underpinned by the recognition that cooperation and conflict are not a binary construct,
as all-or-nothing (Grey and Sadoff, 2002), but rather in co-existence, In their Transboundary
Waters Interaction Nexus (TWINS) tool, Zeitoun and Mirumachi (2008) attended to the nexus of
water conflict and cooperation underlining the dual nature of interaction.
While these inventories provide a global snapshot of conflicts and cooperation in transboundary
rivers, they also provide a simplistic image of the inherent complexity of tensions (De Stefano et
al., 2017). As often stated by Zeitoun and Warner (2006) that "the absence of war does not mean
the absence of conflict". Simply classifying water events into conflict or cooperation could mask
various forms of conflictive or cooperative responses elicited from each riparian state underneath
(Watson et al., 2009). Riparian countries have their respective values and priorities for water
management (Wolf et al., 2005;Di Baldassarre et al., 2013), and their values of shared water often
has possible impacts for their propensity to involve in cooperative management and adhere to
treaties/agreements. Understanding value in the context of transboundary river basins is therefore
vital for developing effective management and policies toward cooperation (Bennett and Dearden,
2014;Hartley, 2006;Larson et al., 2009;Turner et al., 2014).





Values, arising from the concept of culture along with norms and beliefs, posit as deeply held
ideas that influence on water management decisions and outcomes (Caldas et al.,
2015;Roobavannan et al., 2018;Wei et al., 2017). Shaping the way we see, perceive and interpret
the outer environment (Caldas et al., 2015), values is considered as mediating variable that
connect human with the natural environment. In the context of local-scales studies, i.e. urban or
agricultural sectors in river basin (Elshafei et al., 2014;van Emmerik et al., 2014;Li et al.,
2013;Chen et al., 2016;Kandasamy et al., 2014), value is often used synonymously with
"ecological worldviews" or "environmental value" (Schwartz and Bilsky, 1987;White et al., 2019)
that guide water use behavior or management focus from human uses to restore ecological flows
(Roobavannan et al., 2018;Wei et al., 2017). In the context of transboundary rivers, where
multiple water users are interconnected (Petersen-Perlman et al., 2017), their different values
towards their shared water are often manifested as conflictive or cooperative attitudes toward
other competing water users, when further complicated by the interdependent web of
hydrological, political, economic, technical, and social processes (Dinar, 2004;Di Baldassarre et
al., 2019), could resulted in greater cooperation or conflict at basin-scale. Improving
transboundary water management therefore firstly requires nuance understanding the changing
values and interests of each riparian country, however, it remains under researched. Therefore,
current event-based approach is inadequate to recognise the nuance nature of conflict and
cooperation instances. An in-depth analysis that looks into each riparian country's conflictive or
cooperative perspectives is key to understand their cooperative or non-cooperative behaviour.  It
can also provide empirical advances to made it possible more rigorously model social element at
transboundary level in socio-hydrological models or similar studies, and ultimately contribute to
understanding the mechanism that drives the conflict or cooperation choices in the long run. This
paper takes Lancang-Mekong River Basin as a case study to investigate how has the conflict and
cooperation dynamics as reflected by each the riparian countries changed over time.
**2.      Case Study Area - Lancang-Mekong River Basin**
The Lancang-Mekong River is one of the largest and longest river systems in South-East Asia,





originates from the north-eastern rims of the Tibetan plateau in China, down 4,880km through
Myanmar/Burma, Lao PDR (Laos), Thailand, Cambodia and exiting into the South China Sea
through Vietnam (MRC, 2019), as seen in Figure 1. The river courses that runs within China is
named Lancang River, whilst river course flows through downstream is referred as Mekong
River. The Lancang-Mekong River is an essential water source that supports the livelihoods for
some 65 million people from the six riparian countries in maintaining food security and nutrition
(Dugan et al., 2010).
One of the most prominent sources of tension between the riparian states is their competing
desires for the use of the water. China in particular has an interest in hydropower projects to
generate electricity and also in clearing and expanding waterways to improve navigation for
greater trade (Yorth, 2014). Myanmar has access to part of the Lancang-Mekong River through
the sharing of a border with Laos but has not projected a preferred use of the water vocally but is
generally cooperative with China (Yorth, 2014). Laos, similarly to China also has a great interest
in hydropower developments and are in a position favourite to alter the downstream flow of the
Mekong River (Dugan et al., 2010). Thailand primarily utilises water for agriculture and irrigation
and diverts water from the main Lancang-Mekong tributary into its North-eastern areas for
cultivation and exports (Nesbitt, 2005). Cambodia has a particular interest in preserving water
quantity and quality for their fisheries sector to ensure aquatic species abundance (Yorth, 2014).
As a result, Cambodia demands that fewer large structures are constructed along the Lancang-
Mekong, such as dams and irrigation systems that may affect the sediment flow and water quantity
downstream (Yorth, 2014). Vietnam has an interest in utilizing the water for agriculture and
aquaculture and generally contests upstream dams that will have an effect on its  water quantity
for irrigation and aquaculture and its flood control abilities (Nesbitt, 2005).
The Lancang-Mekong River has experienced a lengthy record of conflictive and cooperative
events. Significant movement towards cooperation over water resources between the riparian
countries primarily began in the 1950's when the Mekong Committee was established, consisting
of the lower Mekong countries, after the Geneva Convention granted independence to Laos,
Cambodia, and Vietnam (Hirsch and Cheong, 1996). This committee ran from 1957 to 1978





despite disagreements among the riparian countries in how the decision-making processes were
implemented (Yorth, 2014). In 1995, all members of the Mekong River Commission (hereafter,
MRC ) signed the "Agreement on the Cooperation for the Sustainable Development of the
Mekong River Basin" (Hirsch and Cheong, 1996), with China and Myanmar presenting as
Dialogue Partners of the MRC throughout discussions (Yorth, 2014). The beginning of the 21st
Century has marked China's cooperative commitment for providing 24-hour water level and 12-
hour rainfall data as well as entering cooperative regimes with the MRC (Dore, 2003).
Meanwhile, construction of large-scale dams in upstream has received mounting criticism, i.e.
the Xayaburi Dam, as the first of the eleven proposed cascade dams on the Lower Mekong began
construction in 2010 despite a lack of agreement between all four lower Mekong countries and
failure of the regional consultation process.  After that, several treaties and plans were signed,
including the Lancang-Mekong Environmental Cooperation Centre in 2016 and the formation of
the Lower Mekong Committee (LMC) framework, marking a significant step towards
cooperation.
**3.    Methods**
It has been an important though challenging task to directly measure values related to
environmental concern or specific evaluation towards certain issues (Roobavannan et al., 2018).
News media has increasingly been recognized as a valid proxy to track societal values or public
opinion (Wei et al., 2015;Wei et al., 2017;Quesnel and Ajami, 2017). News media write the first
draft of history (Howland et al., 2006), it provide insights into events reported on by the media
that are representative of each country/sector they are published within (Cooper, 2005). Through
its noted "agenda-setting" capability, news media reflect what is important to the public as well
as it shapes the public perception of an issue (Bengston et al., 1999;Hurlimann and Dolnicar,
2012;Neuendorf, 2017). The prominence of an issue reported in news media can be framed
through frequency of coverage, content details, and prominent position, i.e. front page
(Roznowski, 2003). Recently years have witnessed an increasing trend of examining the water-
related news coverage to understand portrayal of water issues (Altaweel and Bone, 2012;Wei et



al., 2015;Xiong et al., 2016), drought salience (Ruiz Sinoga and León Gross, 2013), public
perception (Hale, 2010), societal values (Wei et al., 2017), or to link the volume of water-related
news coverage with consumption behaviour change (Quesnel and Ajami, 2017) and public
preferences in mitigation strategies (Russell-Verma et al., 2016). Thus, utilizing newspaper
articles as a key data source for this project allows the analysis of the perceptions of different
countries pertaining to water events in the Lancang-Mekong River over time. The approach for
achieving the objectives of this paper is given in Figure 2.
**3.1 Data Retrieval**
The Lexis-Nexis database was selected to extract newspaper information, which is home to more
than 6000 news publications around the world and is among most commonly used news sources
in the field of social sciences (Weaver and Bimber, 2008;Racine et al., 2010). Searching scope
include both major English regional and international newspapers. Although English is not
frequently used in most riparian states, English newspapers are accessible and regularly reach an
international audience, and is therefore considered a reference to the government's foreign policy
(Curtin, 2012). News articles in these newspapers reflect national interests and political responses
that riparian countries want to deliver to the international public.
The search terms are one of the key determinants of the validity and relevance of the data to be
collected. The search terms used in this study, as seen in Table 1, were adopted from Yoffe and
Larson (2001) and refined to enable the results to water events along the Lancang-Mekong river
related to conflict and cooperation between riparian countries. Specifically, the five block of terms
requires articles to be included in the search results must discuss "Mekong" river basin in one of
topics indicated below, such as dam, irrigation, pollution, etc. These articles need to discuss the
conflictive or cooperative aspects of the events involving at least one of riparian countries. The
above categories can narrow down the search to the desired scope, with the list of unwanted words
further screen out irrelevant topics.





### 3.2 Data Cleaning

Initially, the search generated a total of 12,316 results. To further ensure the accuracy of results, all articles were then manually read and examined for their relevancy to ensure the sentiment analysis to be conducted would be reflective of the perspectives of water events along the Lancang-Mekong. Those articles not relevant were removed from the analysis alongside any duplicate articles and those with missing necessary information including article body and date published. The relevancy of each article was determined using the criteria in Table 2. The final number of articles utilized for the analysis was 3,877 after all duplicates and irrelevant articles were removed.

### 3.3 Sentiment analysis and Topic Analysis

Generally, there are two ways of coding available when examining the news content, manual coding and computer-assisted coding. While manual coding could uncover latent content to a larger extent (Wei et al., 2015;Wei et al., 2017), it is more time consuming and less efficient when examining large datasets. Sentiment analysis, a widely used computer-based analysis, was utilized in determining the cooperative or conflictive perspectives towards water events, and how they have changed over time. Sentiment analysis is the process in which thoughts, attitudes and perceptions expressed in a text are identified and classified in computational way, particular in order to determine authors' viewpoints and position towards certain issues (positive, negative, or neutral) (Danneman and Heimann, 2014). The sentiment analysis was conducted through the interface of R, a statistical software program. The process involved inputting textual data into the program, tokenising the sentences to differentiate each word from one another, and then attaching the tokenised text to a sentiment lexicon to identify the overall sentiment (Danneman and Heimann, 2014). As there is no "conflict and cooperation" lexicon for transboundary rivers available, a general sentiment lexicon AFINN was utilized in this analysis. AFINN contains a total of 2,477 attached word-sentiments, which produces a positive and negative value on a scale from -5 to +5 (Nielsen, 2011). In order to represent the conflictive and cooperative sentiments,



the searching scope has limited the articles content to instance of cooperation or conflict that
occurs within an international basin involving one or more riparian to that basin. Therefore, the
calculated sentiment scores based on AFINN scores ranging from - 5 to + 5 was considered being
able to reflect the intensity of conflict and cooperation accordingly.
To reflect topics associated with conflict or cooperation in water events, topic analysis - Structural
Topic Modelling (STM) (Roberts et al., 2014) was utilized. Structural Topic Modelling (STM)
allows frequent words to be extracted from text, identify commonalities between the words,
phrases and groups of words to generate a topical prevalence and topic content factor (Roberts et
al., 2013). This tool is particularly useful when managing big data sources as the process to
identify key topics manually is inefficient and time-consuming, whereas STM has the ability to
identify topics automatically. The STM was processed by using the STM package in R (Roberts
et al., 2014).  The number of topics selected was ten which was decided through an analysis of
the topics produced until clear, relevant topics emerged as a result. For example, at a chosen five
topics, all topics were pertaining to water, resources, and the six riparian countries; however, at
ten topics, there were more clear events emerging such as dam infrastructure, agriculture and
fisheries. The topics were then manually labelled based on the most frequent words found within
each topic as the statistical software cannot extrapolate the overall topic from most frequent
words. This topic classification was based on previous literature reviews and the main water-
related topics outlined in Wei et. al.'s (2015) study.
**4.     Results**
**4.1 Overall Coverage of Conflict/Cooperation Water Events on Lancang-**
**Mekong River Basin**
Overall, news articles pertaining to water conflict/cooperation events along the Lancang-Mekong
River have increased in frequency since 1991. As seen in Figure 3a, Thailand, China and the
international countries consistently have the largest number of articles published each year on this
topic. There are also several peaks in year 2011, 2012, 2014 and 2016 where the number of articles



published were considerably higher than years prior and following.
Overall, conflict and cooperation as reflected in newspaper coverage showed that there was an
observed increase in both the number of conflictive and cooperative articles published over time
(see Figure 3b). From 2014 to 2018, the number of articles with a cooperative sentiment was
more than double that of the number of articles with a conflictive sentiment each year. Number
of cooperative articles had peaked in 2016 and 2018, while definitive peak in the number of
articles published with a conflictive sentiment was in 2011. When examining the relative
prominence of conflictive sentiments to cooperative sentiment over time as seen in Figure 3b,
there has consistently been a greater number of articles with cooperative sentiment than
conflictive sentiments since 2002. This ratio of cooperative to conflictive articles published has
remained relatively stable since 2000 with majority of all years having 60% to 70% of all articles
being cooperative. There are also multiple peaks and troughs in terms of the proportion of
cooperative and conflictive reported articles shown in Figure 3b, peaks were reached in year 2004
and 2015, troughs were found in year 2011.
To understand the most concerned topics associated with conflict/ cooperation events, topic
analysis was conducted with ten topics identified. It was found that nearly one third of all articles
were pertaining to dam infrastructure, implying that this is a significant topic that countries have
a vested interest in (Figure 4a). Following this there is a large proportion of topics that are
associated with the reporting of relationships between countries or their cooperation. 30.4% of
the overall topic proportion includes bilateral relations, multilateral relations, joint management
and meetings. Thus, a significant proportion of all articles published have country interactions
and relationships as a major topic.
Figure 4b also depicts the proportion of topics that are frequently associated with conflictive
sentiment. Dam infrastructure and hydropower, which operate hand-in-hand, were negatively
reported by the media accounting for 60% of the total topics. Whilst another 10% of all negative
articles had a focus on meetings, bilateral relations, flooding and fishing/environment. When
analysing the major topics that prompt a greater cooperative sentiment towards water events in
the media, it is clear that there are five main topics that are focused on: development, meetings,


hydropower, bilateral cooperation and multilateral cooperation (Figure 4c). Development,
meetings and hydropower all are key topics, accounting for 22.22% of topic relevance to articles.
Topic analysis was also conducted in those years that were identified as peaks and troughs. Figure
5 depicts the proportion of topics and most frequent words present in the articles published in
year 2004, 2011 and 2015. 2004 is a year of significance for high proportion of cooperation to
conflict articles published, with approximately 85% of all articles having a positive sentiment. As
per the topic identification and relevance in Figure 5a, main topics reported on during this time
were international relations and bilateral and multilateral projects and cooperation under The
Greater Mekong Subregion (GMS). One major contributor is the Asian Development Bank
(ADB) who provides support to the GMS with the overall objective of poverty reduction, in which
the principal path to this is markets development and cross-border transfer of goods and people
across borders (ADB, 2004). The multitude of annexes and protocols signed throughout 2004 in
regard to a proposed transport facilitation program, with numerous summits and meetings being
held within the riparian countries have contributed to the significantly high proportion of
cooperative sentiment to conflictive sentiment in year. On the other hand, the high proportion of
cooperation score in this year can be also attributed to the absence of often negatively perceived
events, i.e. dam infrastructure.
The year 2011 was a considerable trough as there is a significant drop in the sentiment proportion
with a greater percentage of conflictive articles. This was due to a dramatic increase in the number
of articles published concerning the controversial Xayaburi dam which was identified as one of
the most frequent words in Figure 5b. The major contributing factor of the conflictive sentiments
in this year is the criticism it received from both riparian countries and international community
for the potential impacts of the dam as well as the wrongful consultation process. Finally, in 2015
there was a higher proportion of cooperation articles to conflict. The main topics identified in the
articles published in 2015 are multilateral cooperation and meetings, encompassing four out of
the ten identified topics. This is also corroborated in the word cloud in Figure 5c where the most
frequent words are associated with meetings, development and the inclusion of multiple countries.
In 2015, the early stages of the Lancang-Mekong Cooperation were in development and included





multiple meetings throughout the duration of the year, these meetings are representative of the
countries' movement towards greater cooperation and working towards joint, collaborative
transboundary water management.
**4.2 Conflict and Cooperation Dynamics as Perceived by Each Country**
Sentiment scores for each country was calculated to reveal detailed insights into the evolving
perspectives of each country, as seen in Figure 6a. It was observed that from 1991 to 2018 there
is an apparent trend in increasing cooperative sentiment scores for both international and regional
publications (Figure 6a). There has also been decreased variability in the average sentiments over
time, both international and regional newspaper articles had a similar sentiment score between
2008 and 2018 approximately. The six riparian states have had a greater average sentiment for
cooperation than international countries for the majority of the time scale showing that the region
perceived transboundary water management in the Lancang-Mekong River Basin more positively
than global audiences.
Within the riparian states, upstream riparian countries, such as China, Laos, and Myanmar, are
exhibiting more cooperative sentiments compared to the downstream countries, Cambodia and
Thailand (Figure 6b). However, one major outlier is Vietnam, the most downstream country,
which exhibits the highest sentiment value among all riparian countries. With the exception of
Vietnam's sentiment score, the trend shows that countries further downstream show more
conflictive sentiments. This figure also highlights some of the key players in transboundary river
basin management for the Lancang-Mekong region such as Australia, the United States of
America, and the Philippines. Both Australia and the United States of America are development
partner of the region and thus positively involved in the water management. Philippines is one of
the major publication places for the Asian Development Bank (ADB) which is a key player for
funding and international aid and has been frequently mentioned in the publications.





Most importantly, Figure 7 shows the average sentiment scores for each of the riparian countries
from 1991 until 2018. The results showed that all riparian countries demonstrated mostly
cooperative sentiment relating to water events in Lancang-Mekong River Basin with overall
average sentiments scores from each riparian country in order of lowest to highest are Cambodia
(0.13), Thailand (0.34), Laos (0.46), Myanmar (0.58), China (0.86) and Vietnam (0.91). Upstream
riparian countries, such as China, Laos, and Myanmar, are exhibiting more positive sentiments
compared to the downstream countries, Cambodia and Thailand. However, one major outlier is
Vietnam, the most downstream country, which constantly exhibited the positive sentiment. China
has also consistently expressed very positive sentiments relating to water events in Lancang-
Mekong River Basin over time (Figure7). Upon inspection into the articles from China,
predominantly published by Xinhua News, the Lancang-Mekong Cooperation (LMC) is a
common occurrence in the text that contributes China's positive outlook on transboundary river
basin management in the region. Thailand presents similar results, except for one year, 2011,
which shows a negative average sentiment score. Laos' average sentiment scores between 2007
and 2018 are very variable and do not seem to follow any certain trend. Cambodia showed
predominantly negative average sentiment scores as fishing issues has been a concerned issue
cited in newspaper throughout the study period. Myanmar has minimal data with only 32 articles
were found in total, and only one year, 2014, has shown a negative average sentiment score.
**5.    Discussion and Conclusion**
Understanding value is crucial for establishing effective governance and policies for natural
resources. It is important to understand the change of value toward shared transboundary water
resources, the factors that encourage and discourage changes toward cooperation or conflict. This
paper aimed to develop understanding of the evolution of conflict and cooperation dynamics in
Lancang-Mekong River Basin with in-depth analysis of the perspectives of multiple countries.
Key findings of this study are summarised below.



The overall sentiment analysis in correspondence with the current literature depicts a current trend
of overly cooperative sentiments towards water events occurring within the region. This is
consistent with the previous studies in which the dominant trend in media coverage analysis was
the decreasing of cooperative events from 1948 to 2008 (De Stefano et al., 2010). This research
was also able to bridge the gap in the literature and depict the continual trends that the proportion
of cooperative to conflictive articles has begun to stabilize and started to rise in favour of
cooperative events. There are several reasons for this trend to occur. Firstly, between 1948 and
1999, extensive headway was made towards cooperative actions with the establishment of the
MRC with all lower Mekong countries and the adoption of associated treaties and agreements
throughout its duration (Yorth, 2014). There was also a number of projects in operation outside
the MRC including the "Quadripartite Economic Cooperation (QEC)" with China, Laos,
Myanmar, and Thailand in 1993, the Indicative Basin Plan published in 1970 and the signing of
the agreement on the "Cooperation for the Sustainable Development of the Mekong River Basin"
in 1995 (Yorth, 2014). Moreover, the majority of negative publications are associated with dam
infrastructure and development as per Figure 4, which is also reflective of the worldwide
transboundary rivers with infrastructure and water quantity being identified as key controversial
issues (De Stefano et al., 2010). Therefore, with an absence of dam proposals and construction
prior to the 1990s and hence a significant source of conflict was absent during this time (Yorth,
2014). The general concerns associated with infrastructure development along a river including
limited sediment flow, lower water quality, the effect on fish species and the livelihoods of people
who rely on the river, were not overly present without the threat of infrastructure (Network, 2009).
There is also a likely decline in the percentage of positive articles due to the fact that the Lower
Mekong Basin countries were experiencing civil and regional wars throughout the 1970s to
1980's (Wilson, 2014). As the majority of finances, infrastructure and strategic focus was devoted
to the war during these times, there were no major projects or developments occurring along the
Lancang-Mekong River, contributing to the overall positive perspective of the region.
This study also differentiates between international countries and regional countries in how each
topic is perceived by the media differently, whether riparian is over-critical of water events or





view them from a more cooperative perspective than international countries. This understanding
can allow for greater collaboration in realizing individual concerns of each country and
distributing funding and aid accordingly and ultimately create greater collaborative water
management schemes. It was found that regional countries on average have a higher cooperative
sentiment score than international countries in each year from 1991 to 2018. This is likely
associated with the topics that are considered 'newsworthy' to be published in a regional area,
pertaining to another country. Generally speaking, when countries report on events not occurring
within their close proximity and in different countries, they do so to focus on the major and
complex issues and relationships that occur across the globe (Lewis, 2010). Hence, foreign news
often focuses on significant instances of either great cooperative events such as international
freshwater treaties and major strategic alliances, or significantly conflictive events including
extensive war acts and hostile interactions of both physical and verbal nature (De Stefano et al.,
2010). Given that 38.3 % of the total number of topics reported on are associated with meetings,
bilateral relations, multilateral relations, joint management programs and local water resources as
identified in Figure 6a, it is likely that these topics were not as 'newsworthy' or significantly
cooperative or conflictive enough to be reported on consistently by international countries.
By identifying the perspectives of different types of water events, trends begin to emerge
regarding the frequency of topics resulting in either greater positive or negative sentiments. It was
found that the majority of water events that are negatively reported on are associated with dam
infrastructure (see Figure 4b) and thus, this is likely a major contributor to conflict for the
Lancang-Mekong River Basin. This could be attributed to a variety of reasons. Historically, for
all transboundary river systems, infrastructure and water quantity have been the most contested
events occurring in rivers for their ability to completely alter the current water system and the
significant downstream and upstream impacts (De Stefano et al., 2010). Primarily, major concerns
over the construction of dams is associated with water quantity and the effects this has on
sediment flux changes, water discharge, fisheries and water access for irrigation and agriculture
(Yorth, 2014). Throughout the history of all dam proposals and construction in the Lancang-
Mekong, it is found that not just the construction and operation of the dam that received a





significant amount of negative media attention but also the proposal and planning process.
Therefore, to ensure this pattern of conflict over dam infrastructure is minimized in the future,
investments need to be made in promoting the duty to notify, conducting proper consultation
programs and producing impact assessments available publicly. It was also found that that the
greatest events that are positively reported on by the media are those that aid in connecting leaders
and project developers between riparian countries including meetings, bilateral and multilateral
cooperation and development projects. Development is also generally viewed positively in the
media due to the potential for desired growth and is promoted by many international NGOs
including the ADB. In fact, the ADB aided in the establishment of the Greater Mekong Subregion
Economic Cooperation in 1992 to focus on nine priority areas of economic growth along the
Lancang-Mekong: transport, telecommunications, energy, tourism, human resources
development, environment, agriculture, trade, and investment (Krongkaew, 2004). Thus,
development is considered a crucial topic and action in providing greater cooperation and
collaboration between riparian countries. By allowing this continual interaction and joint projects
that facilitate riparian countries considering all interests and impacts on a larger, transboundary
river scale, there is great potential for future cooperation to solve the current issues within the
Lancang-Mekong Basin.
With the exception of Vietnam's sentiment score, the trend shows that countries further
downstream showed lower positive sentiments. It was predicted that Vietnam and Cambodia
would express negative sentiments, however, these expectations were not met in the study. The
reason behind this pattern is that the true perspectives of some riparian countries including
Vietnam and Cambodia could not be analysed as not many regional newspapers from those
countries were accessible through Lexis-Nexis and as a result hinders the conclusions made. This
is also one of the major limitations of this study that only English newspapers published in
regional and international countries that are accessible through LexisNexis database were
included for analysis.  For future research it is imperative that a greater variety of newspaper
sources covering local languages are utilized through using multiple newspaper databases in order
to gain a representative analysis of the perspectives of all riparian countries.



In conclusion, the future of the Lancang-Mekong is reliant on the riparian countries to
collaboratively manage these resources. If the cooperative water events continue to increase and
the issues associated with negative events can be collaboratively identified, managed and
overcome, there is great potential for the region to achieve effective transboundary water
management. As Kofi Annan, Secretary-General of the United Nations argued in 2002, "… the
water problems of our world need not be only a cause of tension; they can also be a catalyst for
cooperation…If we work together, a secure and sustainable water future can be ours" (Wolf,

452 2007).


**Code/Data availability**
The data is available on request from the corresponding author (tianfq@mail.tsinghua.edu.cn).

**Author contribution**
Jing Wei, Yongping Wei and Fuqiang Tian designed research framework. Jing Wei,
Natalie Nott and Claire de Witt collected data, conducted manual data sorting, and data
analysis. Liying Guo and You Lu revised the code for data analysis. Jing Wei,
Yongping Wei and Fuqiang Tian prepared the manuscripts with contributions from all
co-authors.

**Competing interests**
The authors declare that they have no conflict of interest.

Acknowledgements
We would like to acknowledge the National Key Research and Development Programme of China
(2016YFA0601603) for the funding and support of this research.





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

Hogarth, M., and Dinar, A.: Game Theory and Water Resources Critical Review of its
Contributions, Progress and Remaining Challenges, Foundations and Trends® in
Microeconomics, 11, 1-139, 10.1561/0700000066, 2015.
Howland, D., Becker, M. L., and Prelli, L. J.: Merging content analysis and the policy sciences:
A system to discern policy-specific trends from news media reports, Policy Sciences, 39, 205-
231, 10.1007/s11077-006-9016-5, 2006.
Hurlimann, A., and Dolnicar, S.: Newspaper coverage of water issues in Australia, Water
Research, 46, 6497-6507, https://doi.org/10.1016/j.watres.2012.09.028, 2012.
Kandasamy, J., Sounthararajah, D., Sivabalan, P., Chanan, A., Vigneswaran, S., and Sivapalan,
M.: Socio-hydrologic drivers of the pendulum swing between agricultural development and
environmental health: a case study from Murrumbidgee River basin, Australia, Hydrology and
Earth System Sciences, 2014.
Krongkaew, M.: The development of the Greater Mekong Subregion (GMS): real promise or false
hope?, Journal of Asian Economics, 15, 977-998, 2004.
Larson, K. L., White, D. D., Gober, P., Harlan, S., and Wutich, A.: Divergent perspectives on
water resource sustainability in a public–policy–science context, Environmental Science &
Policy, 12, 1012-1023, 2009.





Lewis, D.: Foreign correspondents in a modern world: The past, present and possible future of
global journalism, The Elon Journal of Undergraduate Research in Communications, 1, 119-127,

557    2010.

Li, J., Dong, S., Peng, M., Yang, Z., Liu, S., Li, X., and Zhao, C.: Effects of damming on the
biological integrity of fish assemblages in the middle Lancang-Mekong River basin, Ecological
Indicators, 34, 94-102, 2013.
Madani, K.: Game theory and water resources, Journal of Hydrology, 381, 225-238,
10.1016/j.jhydrol.2009.11.045, 2010.
McCracken, M., and Wolf, A. T.: Updating the Register of International River Basins of the
world, International Journal of Water Resources Development, 35, 732-782,

565    10.1080/07900627.2019.1572497, 2019.

MRC: State of the Basin Report 2018, VIentiane, Lao PDR, 2019.
Nesbitt, H.: Water used for agriculture in the Lower Mekong Basin, MRC Technical Paper No,

568    11, 1683-1489, 2005.

Mekong    Mainstream    Dams    Threatening    Southeast    Asia's    Food    Security:
https://www.internationalrivers.org/sites/default/files/attached-
files/mekong_mainstream_aug09.pdf, 2009.
Neuendorf, K. A.: The content analysis guidebook second edition, USA: Cleveland State
University, 2017.
Nielsen, F. Å.: A new ANEW: Evaluation of a word list for sentiment analysis in microblogs,
arXiv preprint arXiv:1103.2903, 2011.
Nordås, R., and Gleditsch, N. P.: Climate change and conflict, Political geography, 26, 627-638,

577    2007.

Petersen-Perlman, J. D., Veilleux, J. C., and Wolf, A. T.: International water conflict and
cooperation: challenges and opportunities, Water International, 42, 105-120, 2017.
Petersen‐Perlman, J. D., and Wolf, A. T.: Getting to the first handshake: Enhancing security by
initiating cooperation in transboundary river basins, JAWRA Journal of the American Water
Resources Association, 51, 1688-1707, 2015.



Quesnel, K. J., and Ajami, N. K.: Changes in water consumption linked to heavy news media
coverage of extreme climatic events, Science advances, 3, e1700784, 2017.
Racine, E., Waldman, S., Rosenberg, J., and Illes, J.: Contemporary neuroscience in the media,
Social science & medicine, 71, 725-733, 2010.
Raleigh, C., and Kniveton, D.: Come rain or shine: An analysis of conflict and climate variability
in East Africa, Journal of peace research, 49, 51-64, 2012.
Remans, W.: Water and war, Humantäres Völkerrecht, 8, 1-14, 1995.
Roberts, M. E., Stewart, B. M., Tingley, D., and Airoldi, E. M.: The structural topic model and
applied social science, Advances in neural information processing systems workshop on topic
models: computation, application, and evaluation, 2013, 1-20,
Roberts, M. E., Stewart, B. M., Tingley, D., Lucas, C., Leder‐Luis, J., Gadarian, S. K.,
Albertson, B., and Rand, D. G.: Structural topic models for open‐ended survey responses,
American Journal of Political Science, 58, 1064-1082, 2014.
Roobavannan, M., Van Emmerik, T. H., Elshafei, Y., Kandasamy, J., Sanderson, M. R.,
Vigneswaran, S., Pande, S., and Sivapalan, M.: Norms and values in sociohydrological models,
Hydrology & Earth System Sciences, 22, 2018.
Roznowski, J. L.: A content analysis of mass media stories surrounding the consumer privacy
issue 1990-2001, Journal of Interactive Marketing, 17, 52-69, https://doi.org/10.1002/dir.10054,

601 2003.

Ruiz Sinoga, J. D., and León Gross, T.: Droughts and their social perception in the mass media
(southern Spain), International Journal of Climatology, 33, 709-724, 2013.
Russell-Verma, S., Smith, H. M., and Jeffrey, P.: Public views on drought mitigation: Evidence
from the comments sections of on-line news sources, Urban Water Journal, 13, 454-462, 2016.
Schwartz, S. H., and Bilsky, W.: Toward a universal psychological structure of human values,
Journal of personality and social psychology, 53, 550, 1987.



Song, J., and Whittington, D.: Why have some countries on international rivers been successful
negotiating    treaties?    A    global    perspective,    Water    Resources    Research,    40,
10.1029/2003wr002536, 2004.
Starr, J. R.: Water wars, Foreign policy, 17-36, 1991.
Sunchindah, A.: Lancang-Mekong River Basin: Reflections on cooperation mechanisms
pertaining to a shared watercourse, S. Rajaratnam School of International Studies., 2013.
Tian, F., Liu, H., Hou, S., Li, K., Lu, H., Ni, G., Mu, X., and Baiyinbaoligao: Drought
Characteristics of Lancang-Mekong River Basin and the Impacts of Reservoir Regulation on
Streamflow, Tsinghua University and China Institute of Water Resources and Hydropower
Research, 2020.
Turner, R. A., Fitzsimmons, C., Forster, J., Mahon, R., Peterson, A., and Stead, S. M.: Measuring
good governance for complex ecosystems: perceptions of coral reef-dependent communities in
the Caribbean, Global Environmental Change, 29, 105-117, 2014.
van Emmerik, T. H. M., Li, Z., Sivapalan, M., Pande, S., Kandasamy, J., Savenije, H. H. G.,
Chanan, A., and Vigneswaran, S.: Socio-hydrologic modeling to understand and mediate the
competition for water between agriculture development and environmental health: Murrumbidgee
River basin, Australia, Hydrol. Earth Syst. Sci., 18, 4239-4259, 10.5194/hess-18-4239-2014,

625    2014.

Watson, N., Deeming, H., and Treffny, R.: Beyond Bureaucracy? Assessing Institutional Change
in the Governance of Water in England, Water Alternatives, 2, 2009.
Weaver, D. A., and Bimber, B.: Finding news stories: a comparison of searches using LexisNexis
and Google News, Journalism & Mass Communication Quarterly, 85, 515-530, 2008.
Wei, J., Wei, Y., Western, A., Skinner, D., and Lyle, C.: Evolution of newspaper coverage of
water issues in Australia during 1843–2011, Ambio, 44, 319-331, 2015.
Wei, J., Wei, Y., and Western, A.: Evolution of the societal value of water resources for economic
development versus environmental sustainability in Australia from 1843 to 2011, Global
Environmental Change, 42, 82-92, 2017.



White, D. D., Rauh, E. K., Sullivan, A., Larson, K. L., Wutich, A., Linthicum, D., Horvath, V.,
and Lawless, K. L.: Public attitudes toward urban water sustainability transitions: a multi-city
survey in the western United States, Sustainability Science, 14, 1469-1483, 2019.
Wilson, W. T.: Beating the middle-income trap in Southeast Asia, The Heritage Foundation,
August, 27, 2014.
Wolf, A., Yoffe, S., and Giordano, M.: International Waters: Identifying Basins at Risk, Water
Policy, 5, 29-29, 10.2166/wp.2003.0002, 2003.
Wolf, A. T.: The Transboundary Freshwater Dispute Database Project, Water International, 24,

643    160-163, 10.1080/02508069908692153, 1999.

Wolf, A. T., Natharius, J. A., Danielson, J. J., Ward, B. S., and Pender, J. K.: International river
basins of the world, International Journal of Water Resources Development, 15, 387-427, 1999.
Wolf, A. T., Kramer, A., Carius, A., and Dabelko, G. D.: Managing water conflict and
cooperation, State of the World 2005: redefining global security, 80-95, 2005.
Wolf, A. T.: Shared waters: Conflict and cooperation, Annu. Rev. Environ. Resour., 32, 241-269,

649    2007.

Xiong, Y., Wei, Y., Zhang, Z., and Wei, J.: Evolution of China's water issues as framed in
Chinese mainstream newspaper, Ambio, 45, 241-253, 2016.
CHAPTER 2 BASINS AT RISK: WATER EVENT DATABASE METHODOLOGY:
http://www.transboundarywaters.orst.edu/research/basins_at_risk/bar/BAR_chapter2.pdf 2001.
Yoffe, S., Wolf, A. T., and Giordano, M.: Conflict and cooperation over international freshwater
resources: Indicators of basins at risr 1, JAWRA Journal of the American Water Resources
Association, 39, 1109-1126, 2003.
Yorth, B.: International Mekong River Basin: Events, Conflicts or Cooperation, and Policy
Implications, Master of Public Policy, Oregon State University, 2014.
Zawahri, N. A.: Capturing the nature of cooperation, unstable cooperation and conflict over
international rivers: the story of the Indus, Yarmouk, Euphrates and Tigris rivers, International
Journal of Global Environmental Issues, 8, 286-310, 2008.





Zawahri, N. A., and Mitchell, S. M.: Fragmented governance of international rivers: Negotiating
bilateral versus multilateral treaties, International Studies Quarterly, 55, 835-858, 2011.
Zeitoun, M., and Warner, J.: Hydro-Hegemony - A Framework for Analysis of Trans-Boundary
Water Conflicts, Water Policy, 8, 435-435, 10.2166/wp.2006.054, 2006.
Zeitoun, M., and Mirumachi, N.: Transboundary water interaction I: reconsidering conflict and
cooperation, International Environmental Agreements: Politics, Law and Economics, 8, 297-316,
10.1007/s10784-008-9083-5, 2008.






**List of Figure Captions**

Figure 1.  The location of the Lancang-Mekong River, the main river pathway and its tributaries across the six riparian countries (Tian et al., 2020)

Figure 2.  Outline of the Data Retrieval Process and Coding for Sentiment Analysis and Structural Topic Modelling

Figure 3. The Number of articles published pertaining to water events along the Lancang-Mekong River Basin (a); the proportion of the number of overall positive and negative articles (b)

Figure 4. The proportion of all topics identified as key topics in newspapers from 1991 to 2018 (a); The proportion of Topics Identified within all articles published with an overall conflictive sentiment (b); The proportion of Topics Identified within all articles published with an overall cooperative sentiment (c).

Figure 5. Frequency of Topics identified in all articles published in the year 2004 calculated using STM analysis (a); Frequency of Topics identified in all articles published in the year 2011 calculated using STM analysis (b); Frequency of Topics identified in all articles published in the year 2015 calculated using STM analysis (c)

Figure 6. The Average Sentiment Score of Regional and International Newspapers from 1991 to 2018 (a) and number of articles published relating to water events in the Lancang-Mekong River Basin, average sentiment score for each country (excluding countries with no data), and number of publication sources as denoted by the bubble size (b)

Figure 7. Average sentiment scores for the riparian countries (Cambodia, China, Laos, Myanmar, Thailand, and Vietnam) from 1991 until 2018






700                    Table 1 The Search Terms Established to Generate Results

| Lexis Nexis Requirements | Key Word Search |
|---|---|
| Must Include the words: | Mekong |
| Includes at least one of the following words related to **water**: | water* or river* or lake* or dam* or stream* or tributar* or diversion* or irrigati* or polluti* or "water quality" or flood* or drought* or channel* |
| Includes at least one of the following words related to **conflict/cooperation**: | treat* or agree* or negotiat* or resolution* or commission* or secretariat* or "joint management" or "basin management" or "peace accord" or settle* or cooperat* or collaborat* or dispute* or conflict* or disagree* or sanction* or war* or troop* or "letter of protest" or hostil* or "shots fired" or boycott* or protest* |
| Includes at least one of the following words related to **countries involved**: | Thai* or Cambodia* or China or Chinese or Lao* or Myanmar* or Burm* or "viet nam" or Vietn* |
| Does not include any of the following words: | sea, ocean, navigation, nuclear, "water cannon", "light water reactor", "mineral water", "hold water", "cold water", "hot water", "water canister", "water tight", "water down", "flood of refugees", oil, drugs |






702                           Table 2  Criteria for inclusion and exclusion of news articles

| Criteria for Including Data | Irrigation using the Lancang-Mekong river as a source |
|---|---|
| | Conflict over water resources: e.g. proposed development |
| | Cooperation over water resources: e.g. bilateral/multilateral agreements, MRC, ASEAN |
| | Species affected by development projects: e.g. pollution, water quantity and quality |
| | Salt intrusion due to decreased water quantity and flow from upstream: e.g. dams/diversions |
| | Livelihoods affected by use of water resources: e.g. dams, diversions, dam failures, contamination of water |
| | Flooding or droughts as a result of water release or containment in dams |
| | Infrastructure development that can affect water resources/species e.g. proposed bridge development, dams, diversions |
| Criteria for Excluding Data | Tourism not related to the use of water resources by riparian countries: e.g. cruises, blogs, personal recounts |
| | War: e.g. history of Vietnam War, awarding of medals |
| | Economic development not related to water resources in Lancang-Mekong River |
| | Bridges across the Lancang-Mekong River and not referring to effects on water resources |
| | Tariffs and trade agreements that have no association with water resources |
| | Border conflicts not pertaining to water resources: e.g. security, border control, land ownership disputes |




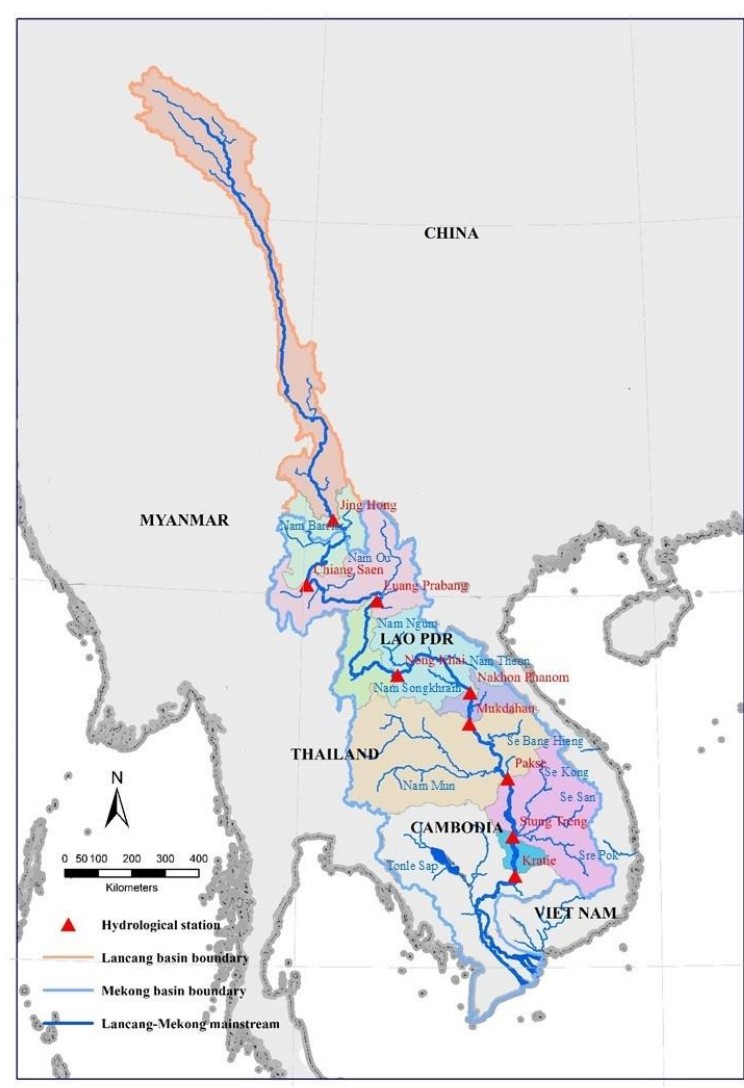


Figure 1   The location of the Lancang-Mekong River, the main river pathway and its tributaries
across the six riparian countries (Tian et al., 2020)



**Step 1**
**Data Retrieval**

Media/Database selection

Search determinants

Data retrieval

12,316 Articles

**Step 2: Manual Data Sorting**

Manually Remove Duplicates and Articles with Missing Information

Remove Irrelevant Articles Based on Pre-determined Criteria

3877 Articles for Coding

**Step 3: Sentiment Analysis**

Step a: Pre-process data through Tokenisation to separate sentences into words and phrases(i.e. also removing stop-words, punctuation and intergers)

Step b: Join each article to sentiment lexicon to identify words associated with positive and negative sentiments (i.e.cooperation and conflict)

Step c: Generate a Data Frame with words from each article associated with a sentiment lexicon and their overall sentiment

3877 Articles with Sentiment Score for Each

Data Visualisation
Excel, Origin, Tableau

**Step 4: Topic Analysis**

Step a: Pre-process data using TextProcessor and readCorpus to remove stopwords, integers, punctuation and refining all words to their roots word through stemming

Step b: Using Structural Topic Modelling Packages, the top 10 main topics were extracted from articles within each year

Step c: Find the proportion of each topic and manually label each topic from the most frequent words generated based on frequent newspaper article topics

Data Visualisation
Excel, Origin, Tableau




Figure 2   Outline of the Data Retrieval Process and Coding for Sentiment Analysis and
Structural Topic Modelling

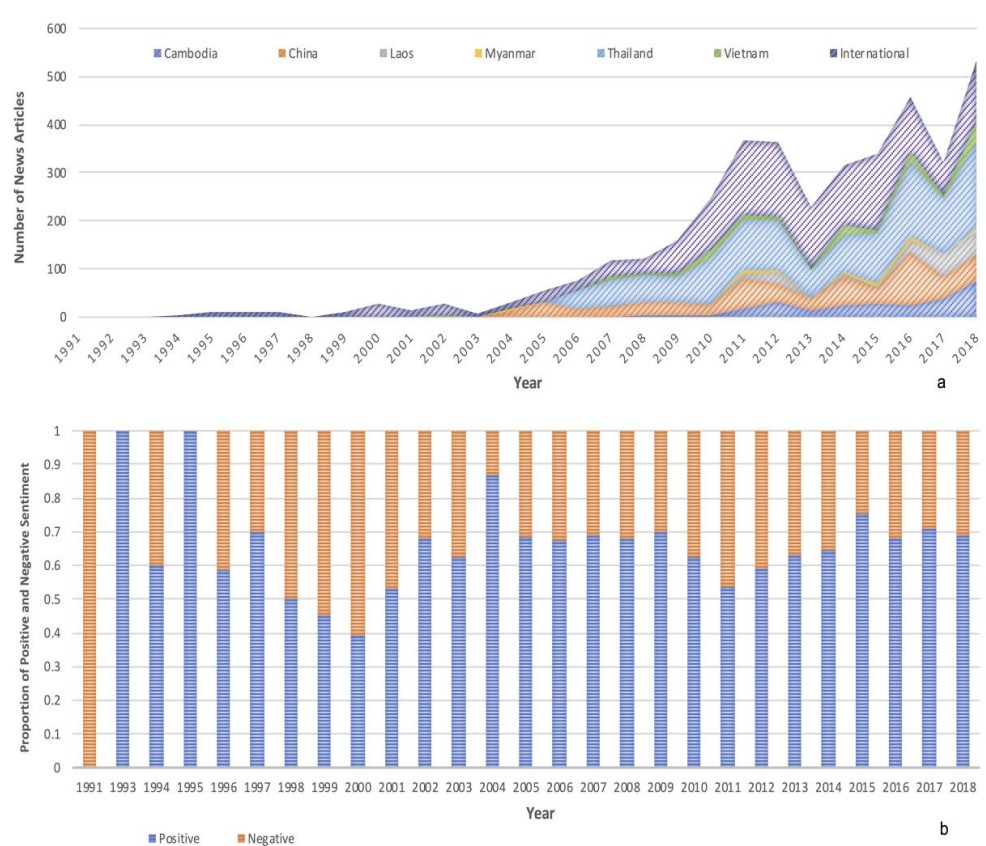

Figure 3. The Number of articles published pertaining to water events along the Lancang-
Mekong River Basin (a); the proportion of the number of overall positive and negative articles
(b)

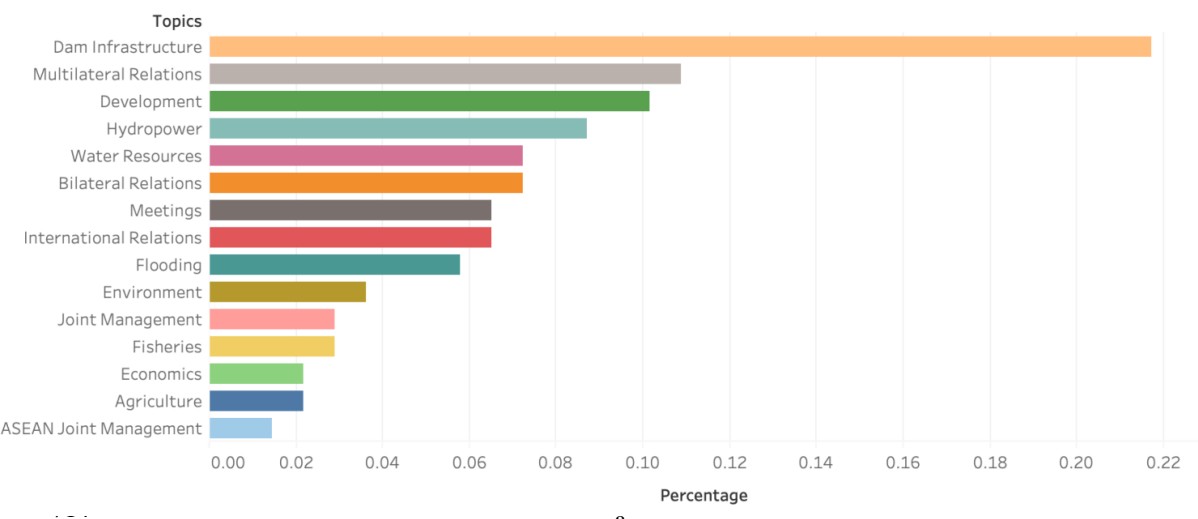

a

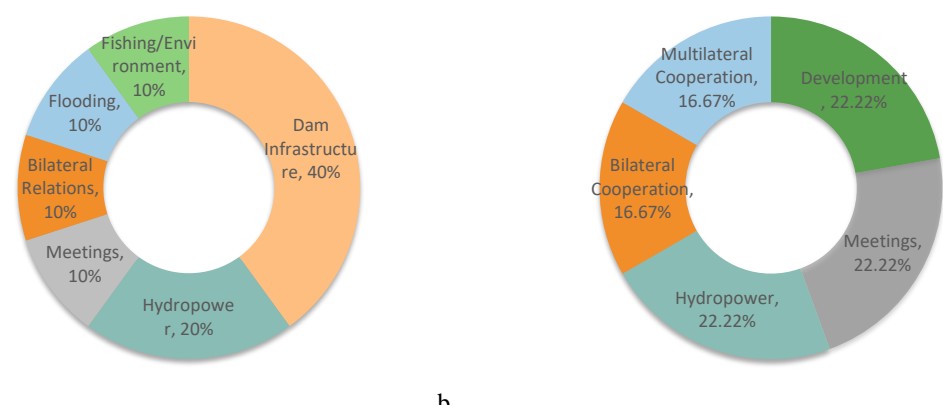

b                                                                    c

Figure 4. The proportion of all topics identified as key topics in newspapers from 1991 to 2018
(a); The proportion of Topics Identified within all articles published with an overall conflictive
sentiment (b); The proportion of Topics Identified within all articles published with an overall
cooperative sentiment (c).

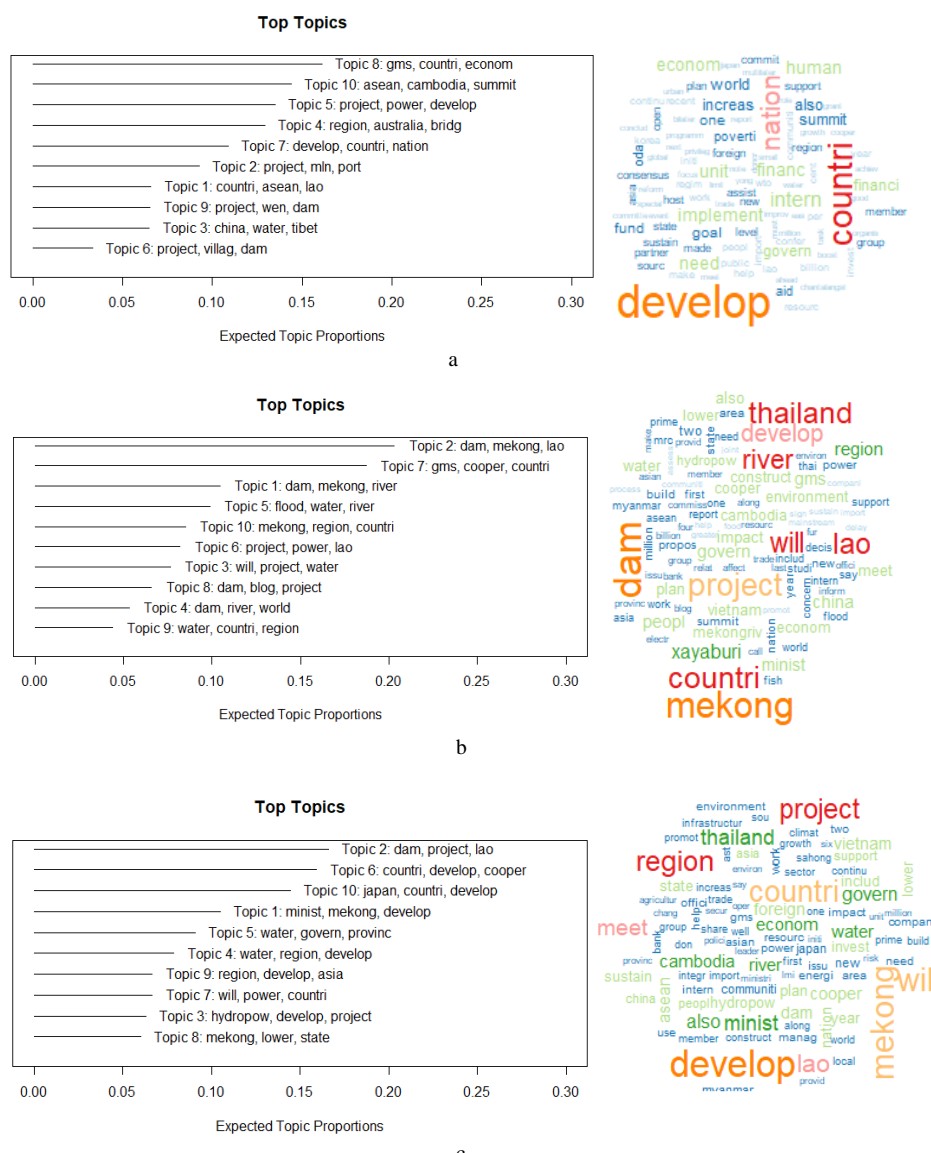



Figure 5. Frequency of Topics identified in all articles published in the year 2004 calculated
using STM analysis (a); Frequency of Topics identified in all articles published in the year 2011
calculated using STM analysis (b); Frequency of Topics identified in all articles published in the
year 2015 calculated using STM analysis (c)





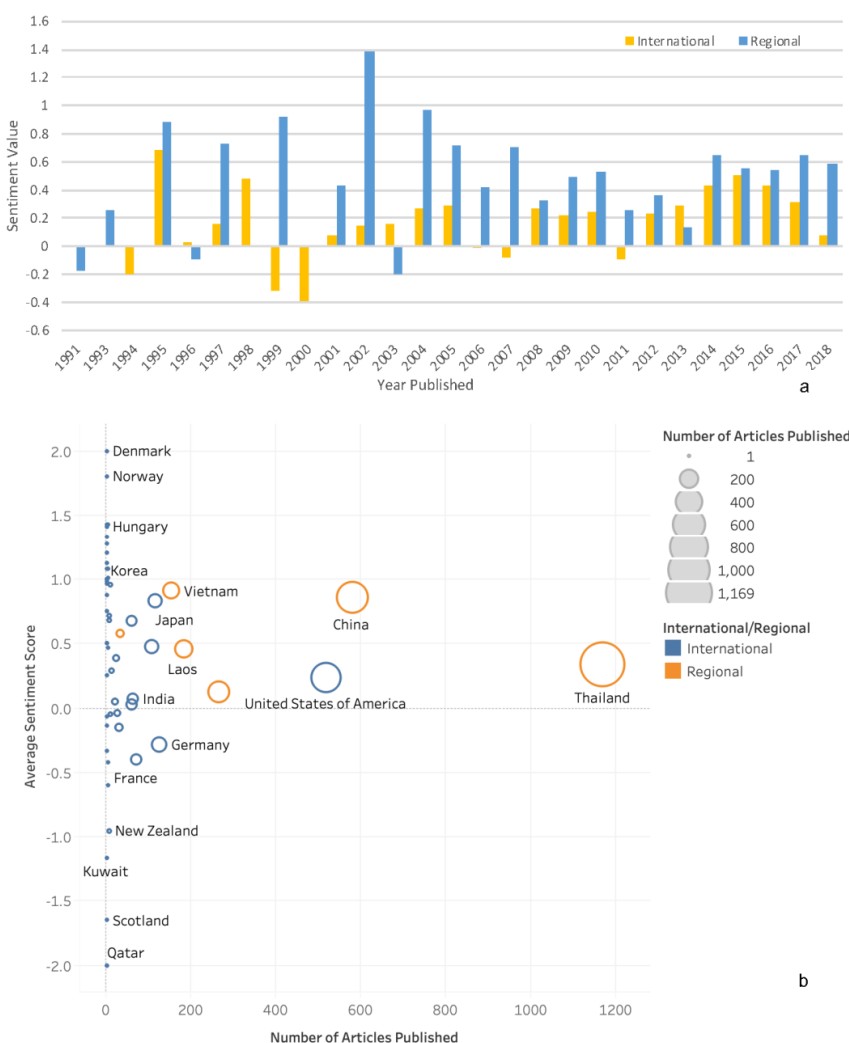

Figure 6. The Average Sentiment Score of Regional and International Newspapers from 1991 to 2018 (a) and number of articles published relating to water events in the Lancang-Mekong River Basin, average sentiment score for each country (excluding countries with no data), and number of publication sources as denoted by the bubble size (b)





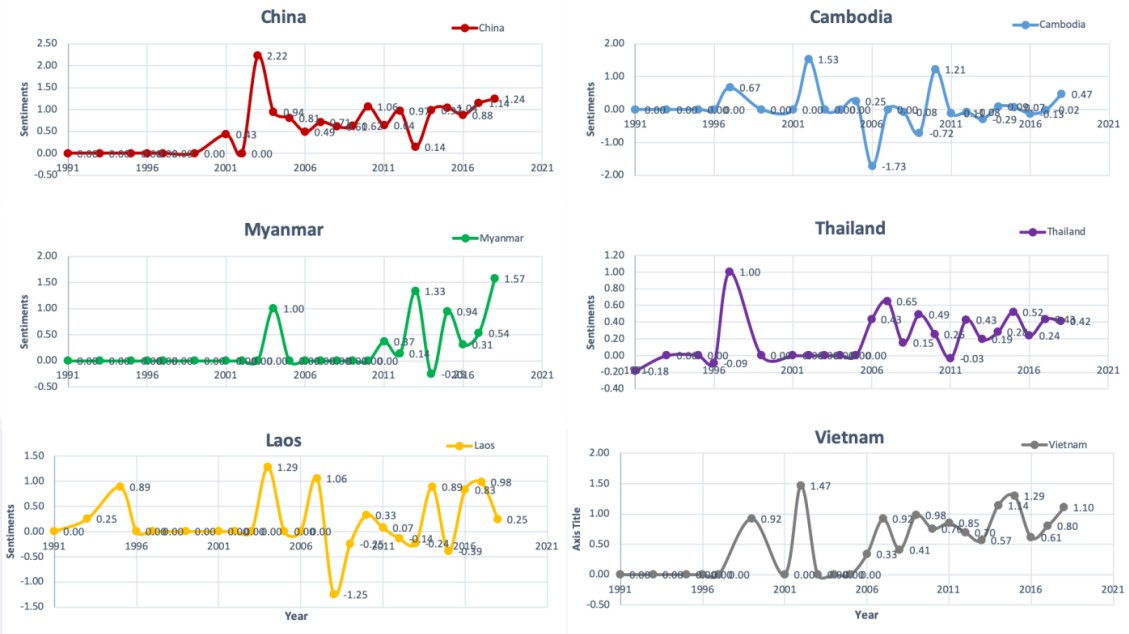

736     Figure 7. Average sentiment scores for the riparian countries (Cambodia, China, Laos,

737     Myanmar, Thailand, and Vietnam) from 1991 until 2018

738