# Peer review of "News Media Coverage of Conflict and Cooperation Dynamics of Water Events in the"

_Hydrology and Earth System Sciences, 2020_

## Referee Comment (RC1) · Anonymous Referee #1 · 1 Oct 2020

This paper presents an analysis of conflict and cooperation sentiments in the Lancang-Mekong river basin countries, for which news media articles were analyzed. The general idea and outcomes are relevant and timely. It is also hopeful that once more it is demonstrated that water more often results in cooperation than in "water wars". However, I think that the paper should be improved substantially before it can be considered for publication in HESS. First, it is not clear what the goal of the paper is. I would strongly encourage to make it very clear in the abstract and introduction what the authors aim to achieve with the paper. As a reader, it makes it sometimes difficult to follow. Second, the general structure may be improved. Since the goal is not directly clear, it is also not logical to first present the case study in detail. I would suggest to make sure

it is clear what research question is being answered, followed by a discussion of the chosen/developed method. Overall, I also think the methods are not presented well. Especially since the chosen methods are perhaps not trivial for the HESS readership, this should be clearly explained earlier in the manuscript. Third, in the introduction a case is made for using socio-hydrology for understanding transboundary river challenges. In the rest of the paper, to link is made to socio-hydrology. I would encourage to make this link in the discussion section. Finally, the manuscript would benefit from a thorough and critical discussion of the methods and results. It is not clear at all what assumptions are made and what uncertainties are introduced in the analysis. In conclusion, the paper has potential, but needs some additional work. Please find below some general comments on all sections, and several specific comments.

**General comments**

**Introduction** The introduction provides a good context for this paper. I do think that the step from water related conflicts and transboundary river basin management to socio-hydrology (SH) is a bit sudden. I suggest to dedicate a short section on introducing SH in this context, and better argue for why SH is a suitable approach. Second, it is not clear to me what the goal of the paper is. Are you presenting a new method? Are you testing a hypothesis? Are you presenting a new perspective? Please make it very clear in the introduction what you aim to achieve with the paper.

**Case study** Nice introduction to the basin. I would suggest to rearrange the sections a bit. Perhaps first discuss the history of events (line 145-163), before discussing the stakes and interests of the individual countries (lines 129-144).

**Methods** The methods should be explained more clearly. The first section of the methods is an overall introduction to "news media" and for the approach the reader is referred to Figure 2 (line 183). Almost eight pages in, it still not clear to me what this paper is about or what method has been developed/used to answer what questions. I suggest to rearrange sections 2 and 3, and start with a very clear description of the goal

of the paper and an overall introduction to the methods. After detailing the components of the method, a description of the case study can be given.

Please provide additional details on how the news article search was conducted. What timeframe was used? What languages of news were considered? What were the generated "results"? Individual newspaper articles or also other media?

As sentiment and topic analysis is not commonly used among the HESS readership, I suggest to maybe illustrate the methods with some examples and even a diagram. This can be in the supplementary materials. For example, show how words and lines are analyzed and what words influence the AFINN score.

**Results** The results are interesting, but read somewhat bulky. Is it possible to present the results in a more concise way? E.g. focus on the main points.

**Discussion and Conclusion** I suggest to use some subheadings, especially since you describe "key points". Would be great to have those points clearly visible as subheadings. Can you elaborate on why dam infrastructure and development is associated with negative sentiment? Can you elaborate on how this cooperation and collaboration has been achieved?

I'm missing a critical evaluation of the chosen method. Are there any sources of uncertainty? Are there any assumptions made about the news media used (e.g. "free journalism" vs "state-run"). What can be improved about the method? Can it be directly applied to other river basins?

**Specific comments** Line 22: Can you elaborate a bit on the method/analysis that was used for this paper?

Line 26: Include the six states.

Line 27: What do you mean with "international countries"?

Line 43: "world's freshwater" –> incorrect, this should be global river flow.

Line 53: Given the somewhat outdated references, do you agree that the idea of "water wars" is also outdated?

Line 67-72: Can you elaborate on why a socio-hydrological approach may be a suitable approach?

Line 76: Does the TFDD include water incidents after 1999? If so, to what date? Can you explain what this -7 to +7 scale is based on and what it means?

Line 86: Omit "often".

Line 88: Can you specify what is defined as "war", "conflict" and "cooperation" in this context?

Line 121-124: Please rephrase, this is a very long sentence.

Line 167: Specify "news media".

Line 185: Provide reference to the Lexis-Nexis database.

Line 189: What do you define as riparian states? What is this claim based on?

Line 185-192: Not clear what languages you included in your analysis. Only English articles?

Line 204: Who read all 12,316 articles? The same person or a team?

Line 206: When was an article considered not relevant?

Line 225: Spell out AFINN.

Line 239: What ten topics were selected?

Line 250: English language articles? What about the overall trend in news articles in the same period? E.g. what is the influence of just overall more news?

Line 354: Why is it important?

Line 388-389: Unclear sentence ("riparian is over-critical"), please rephrase.

Fig. 1: I suggest to also use a background color for the basin in Cambodia and Vietnam. It now looks like the Lancang-Mekong basin stops at Kratie (Cambodia).

Fig. 2: Barely referred to. Either discuss more explicitly or remove.

Fig. 3: Difficult to distinguish the various countries. Try to make the figure clearer, e.g. put the countries on the right side of the figure.

Fig. 4: Very messy figure. Be consistent with percentages (e.g. Fig. 4A has 0-1 scale, unlike Fig. 4B). Also I suggest to only use once decimal for the percentages in Fig. 4B.

Fig. 5: Is there a reason why the words/topics are not spelled out? It reads very messy. (e.g. "countri"). Also I suggest to make it more clear that the three subfigures are for three specific years. E.g. add it to the figures rather than only in the caption.

Fig. 6: Please improve the quality. Specifically the scale of the bubbles in Fig. 6B is not very clear.

Fig. 7: Can you please use the same scale on the y-axis? That would allow a better comparison. Also you can remove the legend in each sub-plot.
* * *

---

## Referee Comment (RC2) · Anonymous Referee #2 · 7 Oct 2020

This paper uses sentiment analysis of newspaper articles to try to develop a narrative around conflict over a transboundary river. The concept of the paper is interesting, novel, and a good contribution to the field of sociohydrology. However, significant revisions are required before the article can be published.

Major Comments

- The findings are presented in a way that is very matter of fact without explicitly telling the reader about the implications of those findings in a broader context.

- The figures need to be significantly improved. There needs to be more attention to detail in the way the figures are presented and they should be restructured such that

the takeaway message from each figure is clearer. Also, please be consistent with colors, fonts, symbols, and wording across all figures to make it easier for the reader to follow along.

Minor Comments

- Title: Consider adding a reference to "news media coverage" or similar in the title to more concisely describe the paper.

- Introduction- The introduction is through, but I would suggest making it more concise and ending the section with a clear description of what you are going to do in the paper, why you are going to do it, how it is novel, and what you hope to glean from the results/what are the implications of the approach

- Line 129- It would be useful to add this information about each country's desired use of water to Figure 1.

- Line 145-163- Adding this timeline as a new figure or to one or more of the existing figures would be helpful for understanding the events that are later reported in the news articles.

- Line 203- This is a lot of newspaper articles, how many people read through these? Did you do any sort of double checking to make sure that people were coding the articles similarly? Also, did you target any specific newspapers or were these global results?

- Line 225- What is AFINN?

- Line 253- Is this because of certain events? There is no mention of specific political actions, events, or climatic episodes that likely influenced news media coverage spikes and inform the article content. - Line 255-267- what do these results mean beyond just the numeric trends?

- Figure 1- I appreciate the map, but this one could be significantly improved for readability.

- Line 319- Interesting takeaway but what does this mean for water resources management?

- Line 327- Good takeaway which could also be communicated on one or more figures by adding upstream -> downstream labels.

- Line 344- Does China's state-controlled media have anything to do with this? Do other countries have state-controlled media as well?

- Line 347- Why is this?

- Line 354- What value?

- Discussion and Conclusion- This section is very long and would benefit from subsections which highlight the key takeaways. Also, what do these results mean beyond the case study region and what are future research directions for this type of news media sociohydrology research?

- Figure 1- I appreciate the map, but it would be more useful if it were annotated and the colors of the countries matched the rest of the figures.

- Figure 2- This is a very useful figure but please reference it more in text. Additionally, you don't need to list the data visualization software that you used.

- Figure 3a- The way this graph is structured, I cannot tell the differences the magnitudes of the categories.

- Figure 3b- What am I supposed to glean from this figure? On line 257 you write that the number of both negative and positive articles has increased, but this figure shows only the percentage of negative vs. positive articles with no discernable trend or context.

- Figure 4b and 4c- Add "conflictive" and "cooperative" to the figures themselves instead

of just the caption and consider using a data visualization other than pie charts which are hard to compare. Also please be consistent with decimal places.

- Figure 5- This is a very hard figure to understand. Why did you choose these years? What do the colors in the word clouds mean? What am I supposed to take away from this figure?

- Figure 6a- You show the average across all newspapers, but it would be more informative to show as boxplots to understand the distributions. Also, please annotate the y-axis to show positive/collaborative vs. negative/conflictive sentiment values.

- Figure 6b- The colors should match those of 6a. Additionally, the figure is hard to read and I am not sure what the takeaway message is from this plot.

- Figure 7- This is very hard to understand. What do the spikes mean? Are there trends I should be noticing? It is challenging to compare the counties as they are on separate figures with different axes.

---

## Editor Comment (EC1) · Murugesu Sivapalan (Editor) · 10 Nov 2020

The paper has benefited from constructive comments from two reviewers who are familiar with the subject matter and the methods used.

Both reviewers are favorably impressed with the topical relevance of the paper, the literature review, the novelty of the methods adopted and the understanding gained from the analyses. The reviewers have identified several shortcomings with the presentation. Addressing these will improve the quality and impact of the paper.

The authors have responded well to these constructive suggestions from the review-

ers. I would like the authors to submit the revised manuscript. Depending on my own reading of the manuscript I might send it back to the two reviewers for a quick appraisal to ensure that they are satisfied with the revisions.

---

## Author Comment (AC1) · 10 Nov 2020

Reviewer #1: This paper presents an analysis of conflict and cooperation sentiments in the Lancang- Mekong river basin countries, for which news media articles were analyzed. The general idea and outcomes are relevant and timely. It is also hopeful that once more it is demonstrated that water more often results in cooperation than in "water wars". How- ever, I think that the paper should be improved substantially before it can be considered for publication in HESS. First, it is not clear what the goal of the paper is. I would strongly encourage to make it very clear in the abstract and introduction what the authors aim to achieve with the paper. As a reader, it makes it sometimes

difficult to follow. Second, the general structure may be improved. Since the goal is not directly clear, it is also not logical to first present the case study in detail. I would suggest to make sure it is clear what research question is being answered, followed by a discussion of the chosen/developed method. Overall, I also think the methods are not presented well. Especially since the chosen methods are perhaps not trivial for the HESS readership, this should be clearly explained earlier in the manuscript. Third, in the introduction a case is made for using socio-hydrology for understanding transboundary river challenges. In the rest of the paper, to link is made to socio-hydrology. I would encourage to make this link in the discussion section. Finally, the manuscript would benefit from a thorough and critical discussion of the methods and results. It is not clear at all what assumptions are made and what uncertainties are introduced in the analysis. In conclusion, the paper has potential, but needs some additional work. Please find below some general comments on all sections, and several specific comments.

-Author response: We would like to thank reviewer #1 for the constructive suggestions and comments, which we believe will help to improve the manuscript substantially. We agree to re-work on the main issues pointed out by the reviewer to progress the manuscript further. Our explanations and responses to all the reviewer's comments and questions are listed below.

General comments Introduction The introduction provides a good context for this paper. I do think that the step from water related conflicts and transboundary river basin management to socio- hydrology (SH) is a bit sudden. I suggest to dedicate a short section on introducing SH in this context, and better argue for why SH is a suitable approach. Second, it is not clear to me what the goal of the paper is. Are you presenting a new method? Are you testing a hypothesis? Are you presenting a new perspective? Please make it very clear in the introduction what you aim to achieve with the paper.

-Author response: We appreciate this suggestion from the reviewer that the link from transboundary river basin management to socio-hydrology should be strengthened and

justified, in the new version, we will elaborate on the socio-hydrological approach, and how this research contribute to more rigorously model social element at transboundary level in socio-hydrological models. The goal of this paper is to provide a new perspective to understand the conflict and cooperation dynamics by highlighting each riparian country's conflictive or cooperative attitude towards their shared water. The justification behind this is riparian countries have their respective values and priorities for water management, their perspective of shared water often has possible impacts for propensity to involve in cooperative management and adhere to treaties/agreements. An in-depth analysis that looks into each riparian country's conflictive or cooperative perspectives is key to understand their cooperative or non-cooperative behavior. We thank the reviewer for raising this concern of confusion, and will rewrite the aim to make it much clearer in the revised manuscript.

Case study Nice introduction to the basin. I would suggest to rearrange the sections a bit. Perhaps first discuss the history of events (line 145-163), before discussing the stakes and interests of the individual countries (lines 129-144).

-Author response: Thank you for this comment. The reason we put the stakes and interests of each riparian country first is to emphasize that it is because of the competing desires for the use of the shared water that have resulted in the following historical conflict and cooperation events. As reviewer remarks below, we will re-arrange the case study and method section.

Methods The methods should be explained more clearly. The first section of the methods is an overall introduction to "news media" and for the approach the reader is referred to Figure 2 (line 183). Almost eight pages in, it still not clear to me what this paper is about or what method has been developed/used to answer what questions. I suggest to rearrange sections 2 and 3, and start with a very clear description of the goal of the paper and an overall introduction to the methods. After detailing the components of the method, a description of the case study can be given.

[Figure]

-Author response: Thank you for this comment. We will re-arrange the structure of this section to have a clear statement of the goal followed by the method and data collection.

Please provide additional details on how the news article search was conducted. What timeframe was used? What languages of news were considered? What were the generated "results"? Individual newspaper articles or also other media?

-Author response: We agree with the specific suggestions and comments. The step-by-step data collection was illustrated in figure 2 as well as in description of line 185-210. We used the keyword search term (table 1) to retrieve articles from the Lexis database, without limiting the timeframe. The language of newspaper was limited to English only. Results generated are individual newspaper articles. In the revised manuscript, we will clarify in more specific details.

As sentiment and topic analysis is not commonly used among the HESS readership, I suggest to maybe illustrate the methods with some examples and even a diagram. This can be in the supplementary materials. For example, show how words and lines are analyzed and what words influence the AFINN score.

-Author response: Yes, we realize that we need to explain the method in more details in this manuscript so that readers who are not familiar with the method can grasp the general idea. As suggested by reviewer, we will put it in supplementary materials with more specific information and illustration.

Results The results are interesting, but read somewhat bulky. Is it possible to present the results in a more concise way? E.g. focus on the main points.

-Author response: We appreciate this suggestion from the reviewer and will incorporate the suggestion and adjust the results section in more concise way.

Discussion and Conclusion: I suggest to use some subheadings, especially since you describe "key points". Would be great to have those points clearly visible as subhead-
ings. Can you elaborate on why dam infrastructure and development is associated with negative sentiment? Can you elaborate on how this cooperation and collaboration has been achieved?

-Author response: We agree with the reviewer and will be more concise in key points. Some sections of the results and discussion will be re- arranged and re-written with the help of this input. As in line 379-381, the general concerns associated with infrastructure development along a river including limited sediment flow, lower water quality, the effect on fish species and the livelihoods of people who rely on the river, were not overly present without the threat of infrastructure. Yes, we can elaborate more on this along with a parallel description of how cooperation can be achieved in revised manuscript.

I'm missing a critical evaluation of the chosen method. Are there any sources of uncertainty? Are there any assumptions made about the news media used (e.g. "free journalism" vs "state-run"). What can be improved about the method? Can it be directly applied to other river basins?

-Author response: The limitation of this study is that only English newspapers were retrieved and included for analysis, which we might miss a variety of local languages newspaper sources that representing the local voices and perspectives. For future research, this could be improved by covering local languages through multiple newspaper databases. To avoid possible media-biases, we did not specify the types of news media to be included (e.g. "free journalism" vs "state-run"). With the keyword search terms, this method can be directly applied to other river basins.

Specific comments Line 22: Can you elaborate a bit on the method/analysis that was used for this paper?

-Author response: Yes, we would add more method/analysis description in the revised manuscript.

Line 26: Include the six states.

-Author response: We will incorporate the suggestion and add the names of six states.

Line 27: What do you mean with "international countries"?

-Author response: This study differentiates between the riparian countries and international countries to reflect the differences of how the water issues are perceived within the basin countries and outside the basin. "International countries" generally referred to all countries outside Mekong River Basin, we will clarify this in the revised manuscript.

Line 43: "world's freshwater" –> incorrect, this should be global river flow.

-Author response: Correction will be made accordingly.

Line 53: Given the somewhat outdated references, do you agree that the idea of "water wars" is also outdated?

-Author response: Given the existing and our research findings, "Water wars" could be an overstatement for majority of the international river basins, and for Mekong as well.

Line 67-72: Can you elaborate on why a socio-hydrological approach may be a suitable approach?

-Author response: Yes, we will incorporate this suggestion in the revised manuscript.

Line 76: Does the TFDD include water incidents after 1999? If so, to what date? Can you explain what this -7 to +7 scale is based on and what it means?

-Author response: The TFDD include water incidents from 1948-1999, then expanded to 2008. We will clarify this in the revised manuscript. The -7 to +7 scales are the intensity of conflict or cooperation events. Negative means conflict, positive score means cooperation.

Line 86: Omit "often".

-Author response: Corrections will be made accordingly.

Line 88: Can you specify what is defined as "war", "conflict" and "cooperation" in this

context?

-Author response: "war" referred to formal declaration of war, whilst "conflict" means varieties of negatively perceived events ranging from mild verbal expression to diplomatic hostile actions or even small-scale military acts, similarly, "cooperation" means range of positive events ranging from minor official exchanges to major strategic alliance.

Line 121-124: Please rephrase, this is a very long sentence.

-Author response: Corrections will be made accordingly.

Line 167: Specify "news media".

-Author response: "news media" in the context of our research referred to traditional newspaper media, other than "social media", clarification will be made in the revised manuscript.

Line 185: Provide reference to the Lexis-Nexis database.

-Author response: Yes, we will incorporate this suggestion in the revised manuscript.

Line 189: What do you define as riparian states? What is this claim based on?

-Author response: Riparian states referred to countries that the shared water flows through. Thank you for the comment, we will clarify this in more specific terms to avoid confusion.

Line 185-192: Not clear what languages you included in your analysis. Only English articles?

-Author response: Yes, the scope of analysis includes English articles only due to language constraints.

Line 204: Who read all 12,316 articles? The same person or a team?

-Author response: The first author and two co-authors manually read through all 12,316 articles. In the initial stages of reading, we randomly select 100 newspaper to read

it independently among three of us. The variability of interpretation of articles was discussed and common definitions were set to ensure consistency and reliability.

Line 206: When was an article considered not relevant?

-Author response: When considering the relevance of an article, an article that is classified as relevant need to discuss the conflictive or cooperative aspects of the events involving at least one of riparian countries. Other than the above criteria, articles that discuss other issues, i.e. tourism, history, were considered as irrelevant.

Line 225: Spell out AFINN.

-Author response: AFINN is the name of the sentiment lexicon consists of a list of English terms rated for valence developed by Finn Årup Nielsen. AFINN is the full name not abbreviation of the lexicon.

Line 239: What ten topics were selected?

-Author response: Structural Topic Modelling (STM) can identify topics automatically. The number of topics it can generate was decided through an analysis of the topics produced until clear, relevant topics emerged as a result. In our case, the number is ten after several round of trials. When we chose five topics, all topics were pertaining to similar topics, i.e. water, resources, and six riparian countries; however, at ten topics, there were more clear events emerging such as dam infrastructure, agriculture and fisheries.

Line 250: English language articles? What about the overall trend in news articles in the same period? E.g. what is the influence of just overall more news?

-Author response: Yes, scope of analysis is English language articles. The overall trend of conflictive and cooperative articles is increased over time, see Figure 3a. To minimize the influence of overall news articles in general in the same period, we looked at relative prominence of conflictive sentiments to cooperative sentiment over time as seen in Figure 3b. From this we will be able to see the ratio between cooperative events

and conflictive events, and the trend of change over time.

Line 354: Why is it important?

-Author response: In the context of transboundary rivers, riparian countries have their respective values and priorities for water management, and their values of shared water often has possible impacts for their propensity to involve in cooperative management and adhere to treaties/agreements. Understanding value is therefore vital for understanding their conflictive/cooperative behavior and developing effective management and policies toward cooperation.

Line 388-389: Unclear sentence ("riparian is over-critical"), please rephrase.

-Author response: Yes, we will revise it in the new version.

Fig. 1: I suggest to also use a background color for the basin in Cambodia and Vietnam. It now looks like the Lancang-Mekong basin stops at Kratie (Cambodia).

-Author response: We appreciate the suggestion and will change the background color for a better illustration.

Fig. 2: Barely referred to. Either discuss more explicitly or remove.

-Author response: Figure 2 is the diagram of how we conduct the method as reviewer pointed out in previous comments. As stated earlier, we will re-arrange the method section along with this Fig to produce a clear description of the method section.

Fig. 3: Difficult to distinguish the various countries. Try to make the figure clearer, e.g. put the countries on the right side of the figure.

-Author response: We appreciate the suggestion, we will adjust the legend and color of each country.

Fig. 4: Very messy figure. Be consistent with percentages (e.g. Fig. 4A has 0-1 scale, unlike Fig. 4B). Also I suggest to only use once decimal for the percentages in Fig. 4B.

-Author response: Thank you for this comment, we will change the percentage in Fig 4B as consistent with Figure 4A

Fig. 5: Is there a reason why the words/topics are not spelled out? It reads very messy. (e.g. "countri"). Also I suggest to make it more clear that the three subfigures are for three specific years. E.g. add it to the figures rather than only in the caption.

-Author response: The word cloud was produced automatically by the frequencies they appear in the articles, the words appeared in the cloud are directly from the original context. Yes, we will add the specific year to the figure itself to make it clearer.

Fig. 6: Please improve the quality. Specifically the scale of the bubbles in Fig. 6B is not very clear.

-Author response: We thank the reviewer for the suggestion, we will improve the layout and scale of the bubbles in revised manuscript.

Fig. 7: Can you please use the same scale on the y-axis? That would allow a better comparison. Also you can remove the legend in each sub-plot.

-Author response: We thank the reviewer for the suggestion, all figures will be updated and improved for a better presentation.
* * *

---

## Author Comment (AC2) · 10 Nov 2020

Reviewer #2: This paper uses sentiment analysis of newspaper articles to try to develop a narrative around conflict over a transboundary river. The concept of the paper is interesting, novel, and a good contribution to the field of sociohydrology. However, significant revisions are required before the article can be published.

-Author response: We appreciate reviewer #2 for his/her constructive suggestions and comments. We agree to re-work on the main issues pointed out by the reviewer to progress the manuscript further. Our explanations and responses to all the reviewer's comments and questions are listed below.

[Figure]

Major Comments - The findings are presented in a way that is very matter of fact without explicitly telling the reader about the implications of those findings in a broader context.

-Author response: We thank the reviewer for this suggestion. We agree to add the implication of the findings in the revised manuscript. The implication of this study can be two-folds, on one hand, by identifying the conflictive/cooperative sentiments exhibited by each country, and the specific topics associated with cooperative/conflictive sentiment, it can serve as a reference for water managers to collaboratively identify, manage and overcome potential conflict to achieve effective transboundary water management; on the other hand, by identifying each countries' conflictive/cooperative perspective, it can also provide empirical advances to more rigorously model social element at transboundary level in socio-hydrological models or similar studies. - The figures need to be significantly improved. There needs to be more attention to detail in the way the figures are presented and they should be restructured such that the takeaway message from each figure is clearer. Also, please be consistent with colors, fonts, symbols, and wording across all figures to make it easier for the reader to follow along.

-Author response: We appreciate this suggestion from the reviewer. We realize that all of our figures need to be improved substantially in terms of the overall presentation quality as also suggested by reviewer #1, we will re-work on this in the revised manuscript.

Minor Comments - Title: Consider adding a reference to "news media coverage" or similar in the title to more concisely describe the paper.

-Author response: Thank you for this comment, yes, we will change our title in a more concise way.

- Introduction- The introduction is through, but I would suggest making it more concise and ending the section with a clear description of what you are going to do in the paper, why you are going to do it, how it is novel, and what you hope to glean from the results/what are the implications of the approach

-Author response: We thank the reviewer for this suggestion, and will rewrite and re-arrange this section to make it clearer in the revised manuscript.

- Line 129- It would be useful to add this information about each country's desired use of water to Figure 1.

-Author response: Yes, we can clarify this in the Figure 1 to emphasize the competing desires of their shared waters.

- Line 145-163- Adding this timeline as a new figure or to one or more of the existing figures would be helpful for understanding the events that are later reported in the news articles.

-Author response:  We agree that a timeline illustrating the historical con-flict/cooperation events would be much helpful for readers to grasp the general history of Mekong River Basin.

- Line 203- This is a lot of newspaper articles, how many people read through these? Did you do any sort of double checking to make sure that people were coding the articles similarly?  Also, did you target any specific newspapers or were these global results?

-Author response: All 12,316 articles were manually read through by the first author and two co-authors.  In the initial stages of reading, we randomly select 100 news-paper to read it independently among three of us.  The variability of interpretation of articles was discussed and common definitions were set to ensure consistency and reliability.  For the target of newspaper, we include global results, but distinguish the sources of newspapers into countries within the basin and outside the basin to reflect the differences of how the water issues are perceived.

- Line 225- What is AFINN?

-Author response:  AFINN is the name of the sentiment lexicon consists of a list of English terms rated for valence developed by Finn Årup Nielsen.

- Line 253- Is this because of certain events? There is no mention of specific political actions, events, or climatic episodes that likely influenced news media coverage spikes and inform the article content.

-Author response: The changes of article number can be attributed to two reasons, one is the nature of news media its self (increase of news coverage in general, climatic episodes, etc), the other is due to certain events. To minimize the influence of news media change in general, we not only looked at the overall number of articles pertaining to conflict or cooperation, but also the relative prominence of conflictive sentiments to cooperative sentiment over time as seen in Figure 3b. From this we will be able to see the trend of change over time.

- Line 255-267- what do these results mean beyond just the numeric trends?

-Author response: The results indicate that Mekong countries have generally showed more cooperative sentiment than conflictive. This is in correspondence with the current literature depicts a continual trend of cooperative events within the region.

- Figure 1- I appreciate the map, but this one could be significantly improved for readability.

-Author response: We appreciate the suggestion and will change the background color for a better illustration.

- Line 319- Interesting takeaway but what does this mean for water resources management?

-Author response: The results showing Mekong countries have generally showed more cooperative sentiment than conflictive. By uncovering topics that are more associated with cooperative sentiment, and topics that are often associated with conflictive sentiment, it can serve as a reference for water managers to collaboratively identify, manage and overcome potential conflict to achieve effective transboundary water management.

- Line 327- Good takeaway which could also be communicated on one or more figures

by adding upstream -> downstream labels.

-Author response: Yes, we agree that one more figure illustrating the spatial difference of their sentiments score would be helpful.

- Line 344- Does China's state-controlled media have anything to do with this? Do other countries have state-controlled media as well?

-Author response: In the data retrieval process, the scope of search was set as English newspapers only, and to avoid possible media-biases, we did not specify the types of news media to be included (e.g. "free journalism" vs "state-run"). As stated in Line 344-346, upon inspection into the articles from China, the positive outlook on transboundary river basin management in the region are predominantly published by state-run media. We did not look in particular for other countries, but we will inspect into this.

- Line 347- Why is this?

-Author response: The reason 2011 is an out-liner as there is a significant drop in the sentiment proportion with a greater percentage of conflictive articles. This was due to a dramatic increase in the number of articles published concerning the controversial Xayaburi dam in Laos.

- Line 354- What value?

-Author response: The term "value" is used synonymously with "ecological worldviews" or "environmental value". We will clarify this term in the revised manuscript to avoid confusion.

- Discussion and Conclusion- This section is very long and would benefit from subsections which highlight the key takeaways. Also, what do these results mean beyond the case study region and what are future research directions for this type of news media sociohydrology research?

-Author response: We agree with the reviewer and will re-arrange this section with key

points in more concise way. As mentioned earlier, this study aims to provide a new perspective of understanding conflict and cooperation dynamic from each individual country's perspective to better understand when and how conflict/cooperation would occur. This method could be applied to other river basins as well to develop nuance understanding of the conflict/cooperation dynamics within the region. This study is limited in examining only English newspaper, which a variety of local voices represented in local languages newspaper sources. For future research, this could be improved by covering local languages through multiple newspaper databases, so that the social element can be more rigorously modeled in socio-hydrological models.

- Figure 1- I appreciate the map, but it would be more useful if it were annotated and the colors of the countries matched the rest of the figures.

-Author response: Yes, we will change the background color for a better illustration.

- Figure 2- This is a very useful figure but please reference it more in text. Additionally, you don't need to list the data visualization software that you used.

-Author response: We thank the reviewer for this comment. Figure 2 is the diagram of how we conduct the method, we will re-arrange the method section along with this Fig to produce a clear description of the method section.

- Figure 3a- The way this graph is structured, I cannot tell the differences the magnitudes of the categories.

-Author response: we will improve this figure for a better illustration.

- Figure 3b- What am I supposed to glean from this figure? On line 257 you write that the number of both negative and positive articles has increased, but this figure shows only the percentage of negative vs. positive articles with no discernable trend or context.

-Author response: On line 257 the number of both negative and positive articles has increased is referred to Figure 3a. As stated earlier, the reason we also looked at

the percentage of negative vs. positive articles (as in Figure 3b) is to minimize the influence of news media change in the same period, we looked at relative prominence of conflictive sentiments to cooperative sentiment over time as seen in Figure 3b. From this we will be able to see the prominence of conflictive versus cooperative sentiment change over time. The results of this trend of change is in correspondence with the current literature depicts a continual trend of cooperative sentiments especially after 1998. We will clarify the layout as well as the description of this section to make it clearer to readers.

- Figure 4b and 4c- Add "conflictive" and "cooperative" to the figures themselves instead of just the caption and consider using a data visualization other than pie charts which are hard to compare. Also please be consistent with decimal places.

-Author response: We appreciate the suggestion, and will update and improve this figure. - Figure 5- This is a very hard figure to understand. Why did you choose these years? What do the colors in the word clouds mean? What am I supposed to take away from this figure? -Author response: The reason why these years were chosen in Figure 5 is stated in line 265-267 as they are the peak and troughs in terms of proportion of cooperative and conflictive articles shown in Figure 3b, peaks in year 2004 and 2015, trough in 2011. The color of word in cloud were generated automatically without any specific meaning. Figure 5 is the main results generated in structural topic modeling analysis, which gives information on what topics are associated with the most conflictive sentiments, and what topics are often associated with more cooperative events. We realize the presentation of this figure could be confusing, we will adjust the layout in revised manuscript for a clear message.

- Figure 6a- You show the average across all newspapers, but it would be more informative to show as boxplots to understand the distributions. Also, please annotate the y-axis to show positive/collaborative vs. negative/conflictive sentiment values.

-Author response: We appreciate the suggestion, and will update and improve this

figure accordingly.

- Figure 6b- The colors should match those of 6a. Additionally, the figure is hard to read and I am not sure what the takeaway message is from this plot.

-Author response: In the results of Figure 6, we are trying to differentiates between international countries and regional countries in how each topic is perceived by the media differently. We appreciate the comment and will improve this figure with better illustration.

- Figure 7- This is very hard to understand. What do the spikes mean? Are there trends I should be noticing? It is challenging to compare the counties as they are on separate figures with different axes.

-Author response: We realize that the message from this figure could not be clear. We will update and improve this figure in revised manuscript.
* * *

---

## Author Response (AR1)

**Final Authors Response**

We would like to thank both Reviewers for their constructive suggestions on how to improve the manuscript. In the following, we provide answers to comments from Referees below. For clarity each answer is structured as follows: (1) RC# comments from Referees, (2) AR# author's response.

**Answers to Comments of Reviewer #1:**

RC#1:This paper presents an analysis of conflict and cooperation sentiments in the Lancang-Mekong river basin countries, for which news media articles were analyzed. The general idea and outcomes are relevant and timely. It is also hopeful that once more it is demonstrated that water more often results in cooperation than in "water wars". How- ever, I think that the paper should be improved substantially before it can be considered for publication in HESS. First, it is not clear what the goal of the paper is. I would strongly encourage to make it very clear in the abstract and introduction what the authors aim to achieve with the paper. As a reader, it makes it sometimes difficult to follow. Second, the general structure may be improved. Since the goal is not directly clear, it is also not logical to first present the case study in detail. I would suggest to make sure it is clear what research question is being answered, followed by a discussion of the chosen/developed method. Overall, I also think the methods are not presented well. Especially since the chosen methods are perhaps not trivial for the HESS readership, this should be clearly explained earlier in the manuscript. Third, in the introduction a case is made for using socio-hydrology for understanding transboundary river challenges. In the rest of the paper, to link is made to socio-hydrology. I would encourage to make this link in the discussion section. Finally, the manuscript would benefit from a thorough and critical discussion of the methods and results. It is not clear at all what assumptions are made and what uncertainties are introduced in the analysis. In conclusion, the paper has potential, but needs some additional work. Please find below some general comments on all sections, and several specific comments.

Authors Response#1: We would like to thank reviewer #1 for the constructive suggestions and comments, which we believe will help to improve the manuscript substantially. We have re-worked on the main issues pointed out by the reviewer to progress the manuscript further. Our explanations and responses to all the reviewer's comments and questions are listed below.

RC#2: The introduction provides a good context for this paper. I do think that the step from water related conflicts and transboundary river basin management to socio- hydrology (SH) is a bit sudden. I suggest to dedicate a short section on introducing SH in this context, and better argue for why SH is a suitable approach. Second, it is not clear to me what the goal of the paper is. Are you presenting a new method? Are you testing a hypothesis? Are you presenting a new perspective? Please make it very clear in the introduction what you aim to achieve with the paper.

Authors Response#2: We appreciate this suggestion from the reviewer that the link from transboundary river basin management to socio-hydrology should be strengthened and justified. The goal of this paper is to provide a new perspective to understand the conflict and cooperation dynamics by highlighting each riparian country's conflictive or cooperative attitude towards their shared water. The justification behind this is riparian countries have their respective values and priorities for water management, their perspective of shared water often has possible impacts for propensity to involve in cooperative management and adhere to treaties/agreements. An in-depth analysis that looks into each riparian country's conflictive or cooperative perspectives is key to understand their cooperative or non-cooperative behavior. We thank the reviewer for raising this concern of confusion, and have rewritten the introduction section to make it clearer in the revised manuscript. A short section dedicated to introduction of socio-hydrology is added in Line 96-108. The goal of the paper is stated in Line 90-95:

RC#3: Case study Nice introduction to the basin. I would suggest to rearrange the sections a bit. Perhaps first discuss the history of events (line 145-163), before discussing the stakes and interests of the individual countries (lines 129-144).

Authors Response#3: Thank you for this comment. The reason we put the stakes and interests of each riparian country first is to emphasize that it is because of the competing desires for the use of the shared water that have resulted in the following historical conflict and cooperation events. As suggested, we have re-arranged the case study section, the history of events are discussed first, followed by discussing the competing interests in Line 207-242.

RC#4: Methods The methods should be explained more clearly. The first section of the methods is an overall introduction to "news media" and for the approach the reader is referred to Figure 2 (line 183). Almost eight pages in, it still not clear to me what this paper is about or what method has been developed/used to answer what questions. I suggest to rearrange sections 2 and 3, and start with a very clear description of the goal of the paper and an overall introduction to the methods. After detailing the components of the method, a description of the case study can be given.

Authors Response #4: Thank you for this comment. We have re-arranged the structure of method section to have a clear statement of the goal followed by description of the step-by-step methods in Line 125-134. As remarks above, the case study section is moved to section 3 followed by the reviewer's suggestion.

RC#5: Please provide additional details on how the news article search was conducted. What timeframe was used? What languages of news were considered? What were the generated "results"? Individual newspaper articles or also other media?

Author response #5: We agree with the specific suggestions and comments. The step-by-step data collection was illustrated in Figure 1 as well as in description of line 127-134 followed by more detailed information in sub-sections 2.1 – 2.3. Overall, we used the keyword search term (table 1) to retrieve articles from the Lexis database, without limiting the timeframe. The language of newspaper was limited to English only. Results generated are individual newspaper articles.

RC#6: As sentiment and topic analysis is not commonly used among the HESS readership, I suggest to maybe illustrate the methods with some examples and even a diagram. This can be in the supplementary materials. For example, show how words and lines are analyzed and what words influence the AFINN score.

Author response #6: Yes, we realize that we need to explain the method in more details in this manuscript so that readers who are not familiar with the method can grasp the general idea. As stated above, we have adjusted the methods section by adding general description of the methods adopted in Line 125-134.

RC#7: **Results** The results are interesting, but read somewhat bulky. Is it possible to present the results in a more concise way? E.g. focus on the main points.

Author response #7: We appreciate this suggestion from the reviewer and have incorporated the suggestion and adjusted the results section by adding one more sub-heading (in Line 255) and re-writing this section in a more concise way.

RC#8: **Discussion and Conclusion:** I suggest to use some subheadings, especially since you describe "key points". Would be great to have those points clearly visible as subheadings. Can you elaborate on why dam infrastructure and development is associated with negative sentiment? Can you elaborate on how this cooperation and collaboration has been achieved?

Author response #8: We agree with the reviewer and will be more concise in key points. With the help of this input, we have re-arranged and rewrite part of the results sections. Meanwhile, the discussion part were carried out following the same key points as in results section. As pointed out by reviewer, the general concerns associated with infrastructure development along a river including limited sediment flow, lower water quality, the effect on fish species and the livelihoods of people who rely on the river, were not present without the threat of infrastructure, as can be seen in Line 332-335.

RC#9: I'm missing a critical evaluation of the chosen method. Are there any sources of un-certainty? Are there any assumptions made about the news media used (e.g. "free journalism" vs "state-run"). What can be improved about the method? Can it be directly applied to other river basins?

Author response #9: The limitation of this study is that only English newspapers were retrieved and included for analysis, which we might miss a variety of local languages newspaper sources that representing the local voices and perspectives. For future research, this could be improved by covering local languages through multiple newspaper databases. To avoid possible media-biases, we did not specify the types of news media to be included (e.g. "free journalism" vs "state-run"). With the keyword search terms, this method can be directly applied to other river basins.

RC#10: **Specific comments** Line 22: Can you elaborate a bit on the method/analysis that was used for this paper?

Author response #10: Yes, as stated in previous responses, we have added more method/analysis description in the revised manuscript in line 125-134.

RC#11: Line 26: Include the six states.

Author response #11: We have incorporated the suggestion and add the names of six states in Line 26.

RC#12: Line 27: What do you mean with "international countries"?

Author response #12: This study differentiates between the riparian countries and international countries to reflect the differences of how the water issues are perceived within the basin countries and outside the basin. "International countries" generally referred to all countries outside Mekong River Basin, we will clarify this in the revised manuscript.

RC#13: Line 43: "world's freshwater" –> incorrect, this should be global river flow.

Author response #13: Correction has been made accordingly.

RC#14: Line 53: Given the somewhat outdated references, do you agree that the idea of "water wars" is also outdated?

Author response #14: Given the existing and our research findings, "Water wars" could be an overstatement for majority of the international river basins, and for Mekong as well. We have re-written part of this section, the statement of "water ways" has been removed.

RC#15: Line 67-72: Can you elaborate on why a socio-hydrological approach may be a suitable approach?

Author response #15: Yes, we have incorporated this suggestion in the revised manuscript in Line 96-102.

RC#16: Line 76: Does the TFDD include water incidents after 1999? If so, to what date? Can you explain what this -7 to +7 scale is based on and what it means?

Author response #16: The TFDD include water incidents from 1948-1999, then expanded to 2008. We have clarified this in the revised manuscript in Line 60. The -7 to +7 scales are the intensity of conflict or cooperation events. Negative means conflict, positive score means cooperation.

RC#17: Line 86: Omit "often".

Author response#17: This whole introduction has been re-arranged and re-written. The original sentence is removed.

RC#18: Line 88: Can you specify what is defined as "war", "conflict" and "cooperation" in this context?

Author response #18: "war" referred to formal declaration of war, whilst "conflict" means varieties of negatively perceived events ranging from mild verbal expression to diplomatic hostile actions or even small-scale military acts, similarly, "cooperation" means range of positive events ranging from minor official exchanges to major strategic alliance. However, the original sentence and paragraph has been re-written, please see Line 67-76.

RC#19: Line 121-124: Please rephrase, this is a very long sentence.

Author response#19: Corrections has been made accordingly in Line 199-201

RC#20: Line 167: Specify "news media".

Author response #20: "news media" in the context of our research referred to traditional newspaper media, other than "social media", clarification has made in the revised manuscript in Line 117, 119 and 121.

RC#21: Line 185: Provide reference to the Lexis-Nexis database.

Author response#21: Yes, we have incorporated this suggestion in the revised manuscript.

RC#22: Line 189: What do you define as riparian states? What is this claim based on?

Author response#22: Riparian states referred to countries that the shared water flows through. Thank you for the comment, we have clarified this in more specific terms to avoid confusion.

RC#23: Line 185-192: Not clear what languages you included in your analysis. Only English articles?

Author response#23: Yes, the scope of analysis includes English articles only due to language constraints as described in Line 138-143.

RC#24: Line 204: Who read all 12,316 articles? The same person or a team?

Author response#24: The first author and two co-authors manually read through all 12,316 articles. In the initial stages of reading, we randomly select 100 newspaper to read it independently among three of us. The variability of interpretation of articles was discussed and common definitions were set to ensure consistency and reliability.

RC#25: Line 206: When was an article considered not relevant?

Author response#25: The criteria of determining whether an article is relevant or not is in Table 2. When considering the relevance of an article, an article that is classified as relevant need to discuss the conflictive or cooperative aspects of the events involving at least one of riparian countries. Other than the above criteria, articles that discuss other issues, i.e. tourism, history, were considered as irrelevant.

RC#26: Line 225: Spell out AFINN.

Author response#26: AFINN is the name of the sentiment lexicon consists of a list of English terms rated for valence developed by Finn Årup Nielsen. AFINN is the full name not abbreviation of the lexicon.

RC#27: Line 239: What ten topics were selected?

Author response#27: Structural Topic Modelling (STM) can identify topics automatically. The number of topics it can generate was decided through an analysis of the topics produced until clear, relevant topics emerged as a result. In our case, the number is ten after several round of trials. When we chose five topics, all topics were pertaining to similar topics, i.e. water, resources, and six riparian countries; however, at ten topics, there were more clear events emerging such as dam infrastructure, agriculture and fisheries.

RC#28: Line 250: English language articles? What about the overall trend in news articles in the same period? E.g. what is the influence of just overall more news?

Author response#28: Yes, scope of analysis is English language articles. We were not able to estimate the overall news articles published in the same period, but to minimize the influence of the evolution of news industry in general, we looked at relative prominence of conflictive sentiments to cooperative sentiment over time as seen in Figure 3. From this we will be able to see the ratio between cooperative events and conflictive events, and the trend of change over time.

RC#29: Line 354: Why is it important?

Author response#29: In the context of transboundary rivers, riparian countries have their respective values and priorities for water management, and their values of shared water often has possible impacts for their propensity to involve in cooperative management and adhere to treaties/agreements. Understanding value is therefore vital for understanding their conflictive/cooperative behavior and developing effective management and policies toward cooperation.

RC#30: Line 388-389: Unclear sentence ("riparian is over-critical"), please rephrase.

Author response#30: Yes, we have revised it in Line 370 – 371: "whether riparian is overly critical of water events or view them from a more cooperative perspective than international countries"

RC#31: Fig. 1: I suggest to also use a background color for the basin in Cambodia and Vietnam. It now looks like the Lancang-Mekong basin stops at Kratie (Cambodia).

Author response#31: We appreciate the suggestion and have changed this figure to have a better illustration. After the re-arrangement of method and case study section, the previous Figure 1 is now Figure 2.

RC#32: Fig. 2: Barely referred to. Either discuss more explicitly or remove.

Author response#32: The previous Figure 2 is the diagram of how we conduct the method as reviewer pointed out in previous comments. As stated earlier, we have re-arranged the method section along with description of the method section in Line 125-134.

RC#33: Fig. 3: Difficult to distinguish the various countries. Try to make the figure clearer, e.g. put the countries on the right side of the figure.

Author response#33: We appreciate the suggestion, we have adjusted the whole results section in a more concise way, the original figure is removed.

RC#34: Fig. 4: Very messy figure. Be consistent with percentages (e.g. Fig. 4A has 0-1 scale, unlike Fig. 4B). Also I suggest to only use once decimal for the percentages in Fig. 4B.

Author response#34: Thank you for this comment, we have changed this figure in a consistent layout as Figure 5 in the revised manuscript.

RC#35: Fig. 5: Is there a reason why the words/topics are not spelled out? It reads very messy. (e.g. "countri"). Also I suggest to make it more clear that the three subfigures are for three specific years. E.g. add it to the figures rather than only in the caption.

Author response#35: The previous word cloud was produced automatically by the frequencies they appear in the articles, the words appeared in the cloud are directly from the original context. However, as stated earlier, we have made major revision in terms of the structure and content of results section, this figure is removed in the revised manuscript.

RC#36: Fig. 6: Please improve the quality. Specifically the scale of the bubbles in Fig. 6B is not very clear.

Author response#36: We thank the reviewer for the suggestion, we have improved this Figure with a different layout and illustration, the updated Figure is in Figure 6b.

RC#37: Can you please use the same scale on the y-axis? That would allow a better comparison. Also you can remove the legend in each sub-plot.

Author response#37: We thank the reviewer for the suggestion, we have updated and improved this figure.

RC#38: This paper uses sentiment analysis of newspaper articles to try to develop a narrative around conflict over a transboundary river. The concept of the paper is interesting, novel, and a good contribution to the field of sociohydrology. However, significant revisions are required before the article can be published.

Author response#38: We appreciate reviewer #2 for his/her constructive suggestions and comments. We agree to re-work on the main issues pointed out by the reviewer to progress the manuscript further. Our explanations and responses to all the reviewer's comments and questions are listed below.

RC#39: The findings are presented in a way that is very matter of fact without explicitly telling the reader about the implications of those findings in a broader context.

Author response#39: We thank the reviewer for this suggestion. The implication of this study can be two-folds, on one hand, by identifying the conflictive/cooperative sentiments exhibited by each country, and the specific topics associated with cooperative/conflictive sentiment, it can serve as a reference for water managers to collaboratively identify, manage and overcome potential conflict to achieve effective transboundary water management; on the other hand, by identifying each countries' conflictive/cooperative perspective, it can also provide empirical advances to more rigorously model social element at transboundary level in socio-hydrological models or similar studies. We have added the statement of implication firstly in Introduction section in Line 108-115, meanwhile, we have re-arranged and adjusted the whole results section to emphasize the linkage of these findings with potential implication.

RC#40: The figures need to be significantly improved. There needs to be more attention to detail in the way the figures are presented and they should be restructured such that the takeaway message from each figure is clearer. Also, please be consistent with colors, fonts, symbols, and wording across all figures to make it easier for the reader to follow along.

Author response#40: We appreciate this suggestion from the reviewer. We have updated all of our figures in terms of the overall presentation quality.

Minor Comments

RC#41: Title: Consider adding a reference to "news media coverage" or similar in the title to more concisely describe the paper.

Author response#41: Thank you for this comment, yes, we have changed our title as "News Media Coverage of Conflict and Cooperation Dynamics of Water Events in the Lancang-Mekong River Basin"

RC#42: Introduction- The introduction is thorough, but I would suggest making it more concise and ending the section with a clear description of what you are going to do in the paper, why you are going to do it, how it is novel, and what you hope to glean from the results/what are the implications of the approach

Author response#42: We thank the reviewer for this suggestion. As responded previously, we have rewritten and re-arranged the introduction section in the revised manuscript.

RC#43: Line 129- It would be useful to add this information about each country's desired use of water to Figure 1.

Author response#43: We appreciate this suggestion. However, we have re-arranged this case study section, with the history of events discussed first, followed by description of the competing interests in Line 207-242. We have also updated the previous Figure 1(current Figure 2), which we afraid will be a bit messy if more texts will be added.

RC#44: Line 145-163- Adding this timeline as a new figure or to one or more of the existing figures would be helpful for understanding the events that are later reported in the news articles.

Author response#44: We agree that a timeline illustrating the historical conflict/cooperation events would be much helpful for readers to grasp the general history of Mekong River Basin, we have added a new Figure 3 in the revised manuscript.

RC#45: Line 203- This is a lot of newspaper articles, how many people read through these? Did you do any sort of double checking to make sure that people were coding the articles similarly? Also, did you target any specific newspapers or were these global results?

Author response#45: All 12,316 articles were manually read through by the first author and two co-authors. In the initial stages of reading, we randomly select 100 newspaper to read it independently among three of us. The variability of interpretation of articles was discussed and common definitions were set to ensure consistency and reliability. For the target of newspaper, we include global results, but distinguish the sources of newspapers into countries within the basin and outside the basin to reflect the differences of how the water issues are perceived.

RC#46: Line 225- What is AFINN?

Author response#46: AFINN is the name of the sentiment lexicon consists of a list of English terms rated for valence developed by Finn Årup Nielsen.

RC#47: Line 253- Is this because of certain events? There is no mention of specific political actions, events, or climatic episodes that likely influenced news media coverage spikes and inform the article content.

Author response#47:The changes of article number can be attributed to two reasons, one is the nature of news media itself (increase of news coverage in general, climatic episodes, etc), the other is due to certain events. To minimize the influence of news media change in general, we not only looked at the overall number of articles pertaining to conflict or cooperation, but also the relative prominence of conflictive sentiments to cooperative sentiment over time as seen in Figure 3. From this we will be able to see the trend of change over time.

RC#48: Line 255-267- what do these results mean beyond just the numeric trends?

Author response#48: The results indicate that Mekong countries have generally showed more cooperative sentiment than conflictive. This is in correspondence with the current literature depicts a continual trend of cooperative events within the region.

RC#49:  Figure 1- I appreciate the map, but this one could be significantly improved for readability.

Author response#49: We appreciate the suggestion and have changed this Figure for a better illustration (current Figure 2).

RC#50: Line 319- Interesting takeaway but what does this mean for water resources management?

Author response#50: The results showing Mekong countries have generally showed more cooperative sentiment than conflictive. By uncovering topics that are more associated with cooperative sentiment, and topics that are often associated with conflictive sentiment, it can serve as a reference for water managers to collaboratively identify, manage and overcome potential conflict to achieve effective transboundary water management.

RC#51: Line 327- Good takeaway which could also be communicated on one or more figures by adding upstream -> downstream labels.

Author response#51: Yes, we agree that one more figure illustrating the spatial difference of their sentiments score would be helpful. We have added Figure 7 for such illustration in the revised manuscript.

RC#52: Line 344- Does China's state-controlled media have anything to do with this? Do other countries have state-controlled media as well?

Author response#52: In the data retrieval process, the scope of search was set as English newspapers only, and to avoid possible media-biases, we did not specify the types of news media to be included (e.g. "free journalism" vs "state-run"). As stated in Line 344-346, upon inspection into the articles from China, the positive outlook on transboundary river basin management in the region are predominantly published by state-run media. We did not look in particular for other countries, but we will inspect into this.

RC#53:  Line 347- Why is this?

Author response#53: The reason 2011 is an out-liner as there is a significant drop in the sentiment proportion with a greater percentage of conflictive articles. This was due to a dramatic increase in the number of articles published concerning the controversial Xayaburi dam in Laos.

RC#54: Line 354- What value?

Author response#54: The term "value" is used synonymously with "ecological worldviews" or "environmental value". We have clarified this term in the revised manuscript to avoid confusion in Line 311.

RC#55:  Discussion and Conclusion- This section is very long and would benefit from subsections which highlight the key takeaways. Also, what do these results mean beyond the case study region and what are future research directions for this type of news media sociohydrology research?

Author response#55: We agree with the reviewer and have re-arranged this section along with results section. The results section are now organized into four subsections, and although the discussion section are without subheadings, it follows the same logic of subsections as results section.  As mentioned earlier, this study aims to provide a new perspective of understanding conflict and cooperation dynamic from each individual country's perspective to better understand when and how conflict/cooperation would occur. This method could be applied to other river basins as well to develop nuance understanding of the conflict/cooperation dynamics within the region. This study is limited in examining only English newspaper, which a variety of local voices represented in local languages newspaper sources. For future research, this could be improved by covering local languages through multiple newspaper databases, so that the social element can be more rigorously modeled in socio-hydrological models.

RC#56:  Figure 1- I appreciate the map, but it would be more useful if it were annotated and the colors of the countries matched the rest of the figures.

Author response#56: We appreciate the suggestion and have changed this Figure for a better illustration (current Figure 2).

RC#57:  Figure 2- This is a very useful figure but please reference it more in text. Additionally, you don't need to list the data visualization software that you used.

Author response#57: We thank the reviewer for this comment. Figure 2 is the diagram of how we conduct the method. As stated previously, we have re-arranged the method section along with this Fig to produce a clear description of the method section.

RC#58:  Figure 3a- The way this graph is structured, I cannot tell the differences the magnitudes of the categories.

Author response#58: we will improve this figure for a better illustration.

RC#59:  Figure 3b- What am I supposed to glean from this figure? On line 257 you write that the number of both negative and positive articles has increased, but this figure shows only the percentage of negative vs. positive articles with no discernable trend or context.

Author response#59: As stated earlier, the reason we also looked at the percentage of negative vs. positive articles (as in current Figure 4) is to minimize the influence of news media change in the same period. From this we will be able to see the prominence of conflictive versus cooperative sentiment change over time. The results of this trend of change is in correspondence with the current literature depicts a continual trend of cooperative sentiments especially after 1998. In the revised manuscript, we have rewritten this part and removed the previously Figure 3a to make it clearer to readers.

RC#60: Figure 4b and 4c- Add "conflictive" and "cooperative" to the figures themselves instead of just the caption and consider using a data visualization other than pie charts which are hard to compare. Also please be consistent with decimal places.

Author response#60:We appreciate the suggestion, this figure is now updated as Figure 5b and Figure 5c in the revised manuscript.

RC#61: Figure 5- This is a very hard figure to understand. Why did you choose these years? What do the colors in the word clouds mean? What am I supposed to take away from this figure?

Author response#61: We realize the presentation of this figure could be confusing, in the revised manuscript, we have removed this figure, re-arranged and re-written the results section.

RC#62: You show the average across all newspapers, but it would be more informative to show as boxplots to understand the distributions. Also, please annotate the y-axis to show positive/collaborative vs. negative/conflictive sentiment values.

Author response#62: We appreciate the suggestion, we have updated this Figure as current Figure 7a and Figure 7b.

RC#63: Figure 6b- The colors should match those of 6a. Additionally, the figure is hard to read and I am not sure what the takeaway message is from this plot.

Author response#63: In the results of previous Figure 6, we are trying to differentiates between international countries and regional countries in how each topic is perceived by the media differently. We appreciate the comment and have improved this figure with better illustration as Figure 7a and Figure 7b.

RC#64: Figure 7- This is very hard to understand. What do the spikes mean? Are there trends I should be noticing? It is challenging to compare the counties as they are on separate figures with different axes.

Author response#64: We realize that the message from this figure is not clear. we have updated and improved this figure as Figure 8 with all countries ploted in the same Y axis.

---

## Author Response (AR2)

**Reviewer Comments**

We would like to thank both Reviewers for their positive reviews and further comments to improve this manuscript in technical details. In the following, we provide answers to comments from Referees below. In addition, we have also improved this version of the manuscript in general with track changes. For clarity each answer is structured as follows: (1) RC# comments from Referees, (2) AR# author's response.

**Answers to Comments of Reviewer #1:**

RC#1: The paper is significantly improved and I applaud the authors' efforts to respond to reviewer feedback. In particular, the introduction and framing are significantly improved to describe the motivation and novelty of the work. The figures are also much clearer and cohesive. I suggest the paper be published with only very minor revisions to a few of the figures.

Authors Response#1: we appreciate the reviewer for the positive feedback, and have revised our manuscript in respond to reviewer's suggestion.

RC#2: Figure 3- what do the colors on the timeline mean?

Authors Response#2: we thank the reviewer for raising this question. The different colors denote events in different nature, "orange color" refers to interruption of the Mekong Committee, "blue color" refers to projects and "green color" refers to policy/strategy, we have clarified this in the revised manuscript Line 879-881

RC#3: Figure 6b- consider changing the color scale to be divergent with white at 0 to convey negative vs. positive sentiment

Authors Response#3: Corresponding changes has been made to Figure 6b for a clear presentation of negative vs. positive sentiment.

RC#4: Figure 7- The legend says that the sentiment score can swing between -5 and 5, but then the legend only shows colors for -1 to 1 and the map only shows positive values.

Authors Response#4: Yes, the sentiment score can swing between -5 to +5, however, most riparian countries exhibit moderate sentiment value, which is between -1 and 1.

**Answers to Comments of Reviewer #2:**

Some final (technical) comments:
RC#5: Split discussion and conclusions. No reason to combine. Flow of the story may be improved and main findings will be more clearly communicated.

Authors Response#5: We thank the reviewer for this suggestion. In the revised manuscript, we have addressed this suggestion to have a split discussion and conclusion section.

RC#6: Data availability statement should be updated to be accordance with HESS policy. If data are not openly available, an explicit statement should be made why this is not the case (https://www.hydrology-and-earth-system-sciences.net/policies/data_policy.html).

Authors Response#6: As per the suggestion from reviewer, we have update our data availability statement.

RC#7: Fig. 4 and 6: 1992 is missing so maybe update x-axis. If no data are available also include but leave out the bar.

Authors Response#7: Both Figure 4 and 6a has been updated in the revised manuscript.

RC#8:  Fig. 5: update x-axis, either use percentages or proportion, but not both.

Authors Response#8: We thank the reviewer for this specific suggestion, and have updated x-axis in Figure 5.

RC#9:  Fig. 7: Are there no tributaries between the Laos-China border and the most upstream location? Also the most upstream tributaries are not clear.

Authors Response#9: Yes, as shown in the map, there are almost no major tributaries in the most upstream location, most tributaries are located downstream.

 RC#10:  Fig. 8: Please don't use standard excel "smoothened graph". Connect the dots linearly or explain why you use some polynomial interpolation between the datapoints.

Authors Response#10: We thank the reviewer for this suggestion, Figure 8 has been updated.